# A terahertz meta-sensor array for 2D strain mapping

Xueguang Lu[1,9], Feilong Zhang [2,3,9], Liguo Zhu[4], Shan Peng[1], Jiazhen Yan[5], Qiwu Shi [1], Kefan Chen[1], Xue Chang[1], Hongfu Zhu[1], Cheng Zhang [6,7] ✉, Wanxia Huang [1] ✉ & Qiang Cheng [8] ✉

Large-scale stretchable strain sensor arrays capable of mapping two-dimensional strain distributions have gained interest for applications as wearable devices and relating to the Internet of Things. However, existing strain sensor arrays are usually unable to achieve accurate directional recognition and experience a trade-off between high sensing resolution and large area detection. Here, based on classical Mie resonance, we report a flexible meta-sensor array that can detect the in-plane direction and magnitude of preloaded strains by referencing a dynamically transmitted terahertz (THz) signal. By building a one-to-one correspondence between the intrinsic electrical/magnetic dipole resonance frequency and the horizontal/perpendicular tension level, arbitrary strain information across the meta-sensor array is accurately detected and quantified using a THz scanning setup. Particularly, with a simple preparation process of micro template-assisted assembly, this meta-sensor array offers ultrahigh sensor density (~11.1 cm$^{-2}$) and has been seamlessly extended to a record-breaking size (110 × 130 mm$^2$), demonstrating its promise in real-life applications.

Stretchable strain sensors serving as a bridge between the mechanical and digital worlds, play a key role in the perception layer of applications such as the Internet of Things (IoTs), wearable electronics and soft robotics[1–7]. In these scenarios, large-area flexible sensor arrays with anisotropic nodes are urgently required to map actual deformation, which considers the space-variant and complex-multiaxial features of real-life strain distributions.

Traditional strain sensor arrays based on resistive[8–18], capacitive[19,20] and piezoelectric[21,22] effects integrate strain-sensitive structures, stretchable electrodes, and interconnection materials in a tight space[23], thereby suffering from limited scalability and detection

resolution. Although novel field effect transistor-based sensing networks can significantly improve the perceptual resolution due to their unparalleled device density[16,24], the complex manufacturing processes pose an obstacle for necessary large-area and practical applications. In addition to considering the relationship between the compatible array area and sensor density, multidirectional strain sensing ability should be merged into strain sensor arrays to adapt to various surface strain environments. However, existing anisotropic strain sensors mainly focus on discrete point strain detection as achieved by structural design[25,26] (e.g., cross-strain sensors) and material optimization[27–29] (e.g., anisotropic conductive materials). For the arrays of these

[1]College of Materials Science and Engineering, Sichuan University, Chengdu 610065 Sichuan, China. [2]CAS Key Laboratory of Bio-inspired Materials and Interfacial Science, CAS Center for Excellence in Nanoscience, Technical Institute of Physics and Chemistry, Chinese Academy of Sciences, 100190 Beijing, China. [3]Center for Flexible Devices (iFLEX), School of Materials Science and Engineering, Nanyang Technological University, Singapore 639798, Singapore. [4]Institute of Fluid Physics, China Academy of Engineering Physics, Mianyang 621900 Sichuan, China. [5]School of Mechanical Engineering, Sichuan University, Chengdu 610065 Sichuan, China. [6]Key Laboratory of Materials for High-Power Laser, Shanghai Institute of Optics and Fine Mechanics, Chinese Academy of Sciences, Shanghai 201800, China. [7]Hangzhou Institute for Advanced Study, University of Chinese Academy of Sciences, Hangzhou 310024, China. [8]Department of Radio Engineering, State Key Laboratory of Millimeter Waves, Southeast University, Nanjing 210096, China. [9]These authors contributed equally: Xueguang Lu, Feilong Zhang. ✉e-mail: czhangseu@foxmail.com; huangwanxia@scu.edu.cn; qiangcheng@seu.edu.cn

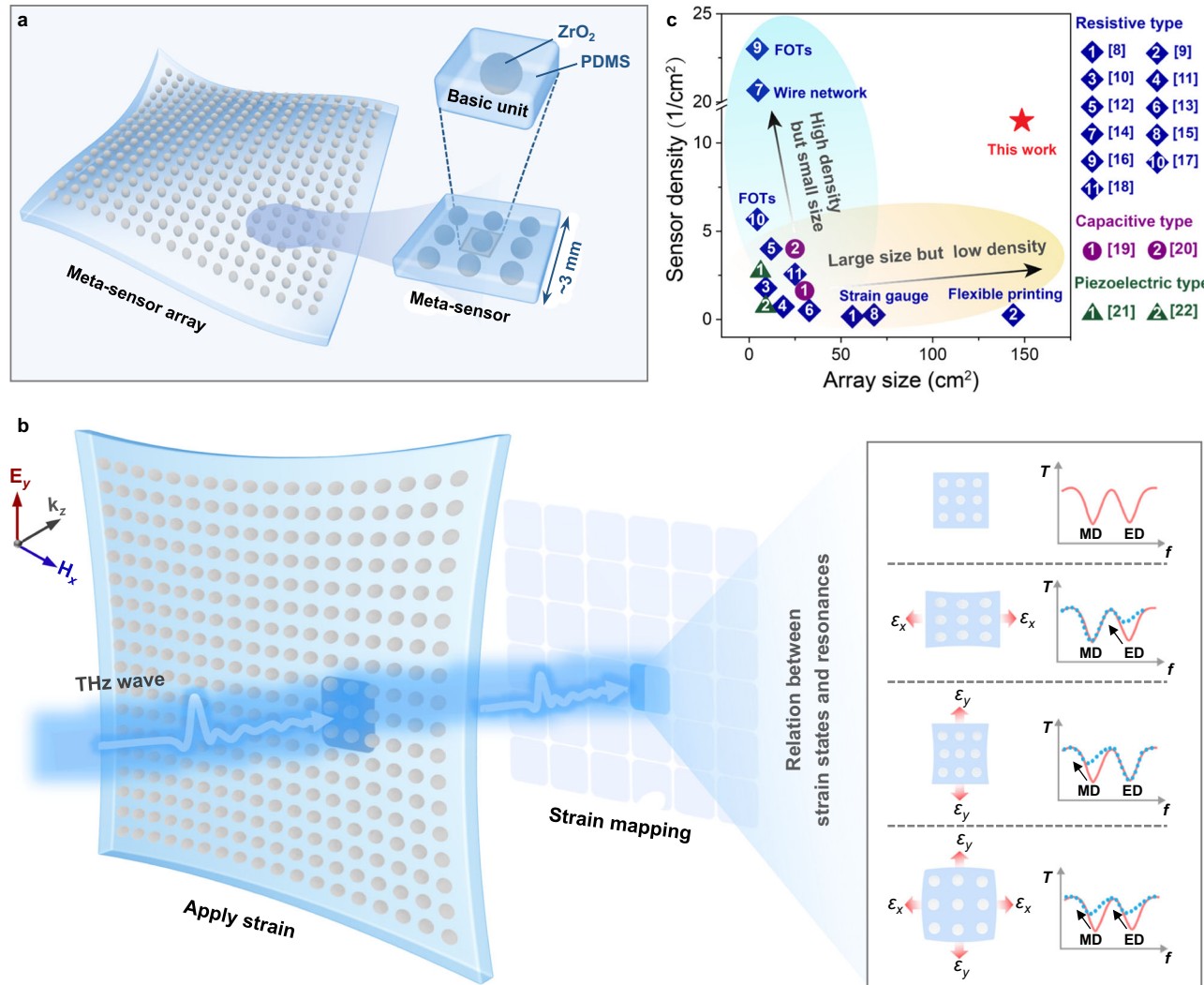

**Fig. 1 | Artistic rendering of a meta-sensor array.** Schematic diagram of the proposed stretchable meta-sensor array (**a**) and its detection principle (**b**). The meta-sensor node and its basic unit cell are enlarged in the inset of **a**. The relationship between the applied strain states (including amplitude and direction) and the ED/MD resonance excited by the THz wave are supplied in the inset of **b**. **c** Array size and sensor density chart of the existing sensor arrays (plotted in different colors according to different working mechanisms), with this work presented as a red star (detailed comparisons of the references in **c** with our design can be found in Supplementary Table S1).

sensors, precise calibration and multidimensional signal decoupling need to be preferentially addressed to overcome prevailing challenges[6,30]. On the other hand, stretchable metamaterials, as an alternative strategy, possess inherent advantage in constructing continuously large-scale sensor array and sensors based on polarization-dependent resonance-strain response have been achieved by patterning metal films on elastomer substrates[31–33]. However, limited by the inevitable correlative deformations in metallic patterns (coupled *x*- and *y*-directional deformations), they are only equipped with small ultimate-strain sensing capability (up to 6%) and powerless in strain direction recognition. In addition, due to the limited interface bonding and mismatched mechanical properties, metallic patterns in existing plasmonic metamaterials tend to be damaged or even peel off the substrate after massive cyclic stretching, thereby constraining their overall performance and durability.

Herein, a Mie resonance-based THz meta-sensor is proposed to detect arbitrary-directional strain in a two-dimensional (2D) plane using the deep decoupling feature between the plane-wave excited electrical dipole (ED) and magnetic dipole (MD) resonances[34–36], which are independent to the *x*- and *y*-directional deformations, respectively. Our meta-sensor array is constructed by encapsulating a high-

permittivity zirconia ($ZrO_2$) microsphere array in a polydimethylsiloxane (PDMS) substrate (Fig. 1a), and its size can reach $110 \times 130\ mm^2$ (containing ~1580 meta-sensors, each of which has a size of ~$3 \times 3\ mm^2$) via a simple micro template-assisted assembly strategy. Combined with existing THz scanning technology (Fig. 1b), applied biaxial strain in a large area are mapped with a high spatial resolution (~3 mm) as a result of the ultrashort wavelength of the incident THz waves (e.g., ~0.7 mm at 0.4215 THz – the initial ED resonance frequency and ~0.9 mm at 0.3220 THz – the initial MD resonance frequency). Comprehensively considering the device size and sensing resolution (Fig. 1c and Supplementary Table S1), this flexible strain meta-sensor array shows great potential for future applications in the sensing layer of flexible IoTs and wearable devices.

## Results and discussion

### Bidirectional strain detection strategy

Classical Mie resonance theory indicates that by deliberately tuning the permittivity (much larger than that of its surroundings) and diameter of a dielectric microsphere, a couple of MD and ED resonances can be generated at different frequencies when interacting with specified normal incident waves, resulting in the magnetic field (H-field)

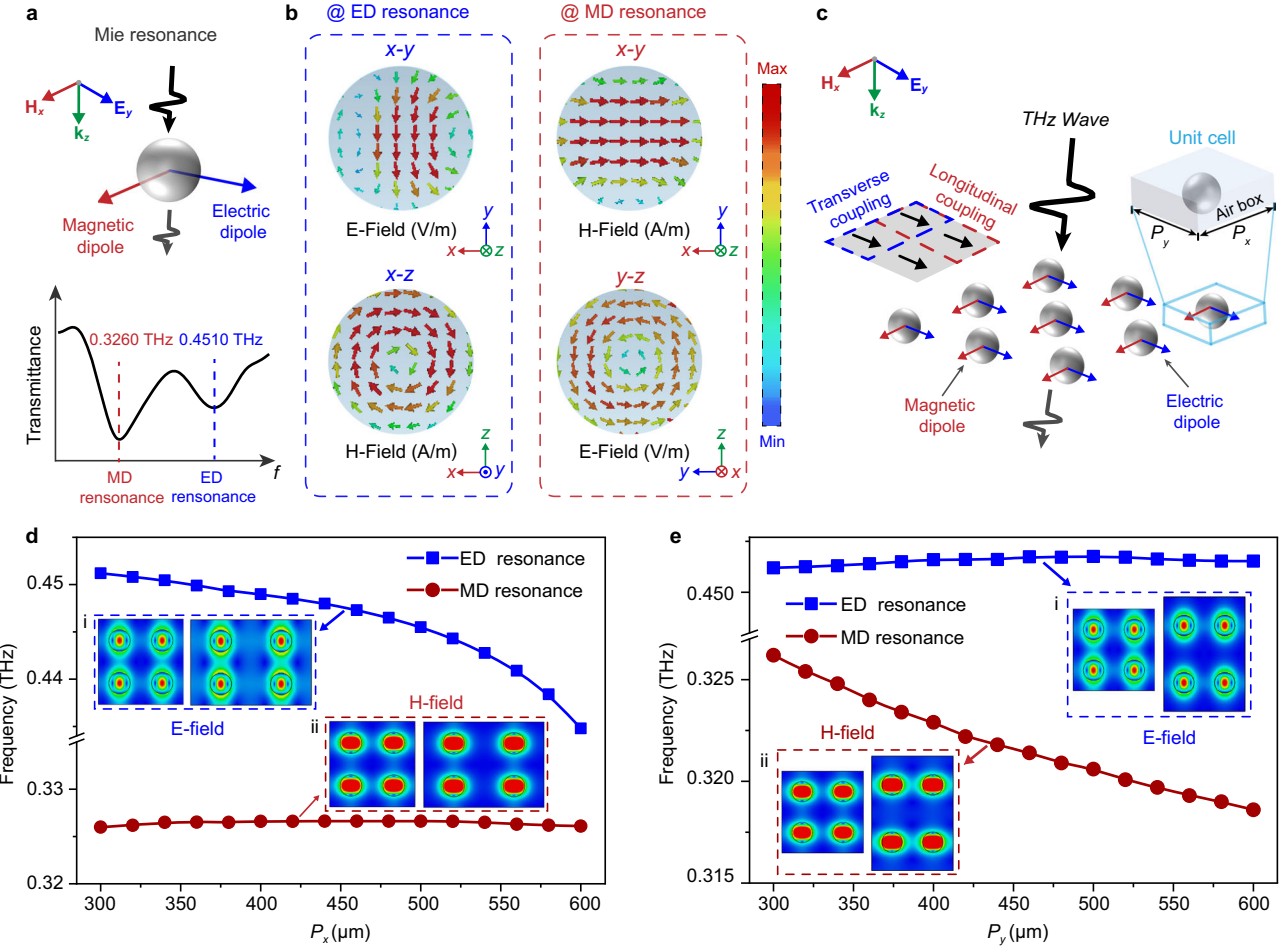

**Fig. 2 | Analytical study of Mie resonance and its array. a** Schematic diagram of the intrinsic ED and MD resonances of a dielectric microsphere (with a dielectric constant of 33 and a diameter of 160 μm) excited by THz waves (top) and its simulated result (bottom). **b** Simulated E-field and H-field distributions at the center cross section of the microsphere regarding the ED (left) and MD (right) resonances. **c** Schematic diagram of the ED and MD distribution in the array of dielectric microspheres. The inset on the left shows two coupling modes: transverse coupling (side-by-side alignment in blue dashed square) and longitudinal coupling (end-to-

end alignment in red dashed square) between the dipoles. The inset on the right illustrates the unit cell parameters ($P_x = P_y = 300$ μm) and the boundary condition. Simulated resonance frequencies (including ED and MD resonances) versus cell period $P_x$ (**d**) and $P_y$ (**e**). The insets of (**d**) and (**e**) present the simulated E-field (i) and H-field (ii) distributions across the center *x-y* plane of the microspheres as $P_x$ (**d**)/$P_y$ (**e**) = 300 μm (left side of the dashed square) and $P_x$ (**d**)/$P_y$ (**e**) = 460 μm (right side of the dashed square).

and electric field (E-field) being strongly localized in its center[37,38] (Supplementary Note S1). This orthogonal field response may be regarded as the basis for bidirectional strain detection. Based on corresponding Mie theory, an ED resonance at 0.4510 THz and an MD resonance at 0.3260 THz are achieved using a well-selected microsphere (Fig. 2a) with a dielectric constant of 33 and a diameter of 160 μm.

To investigate the physical mechanism involved, the near-field distribution inside the microsphere is investigated using CST Microwave Studio 2021 (Fig. 2b). For the ED resonance, the E-field oscillating along the *y*-direction is strongly localized at the center of the *x-y* plane (upper left of Fig. 2b), and the orthogonal H-field with similar behavior concentrates at the identical cutting plane regarding the MD resonance (upper right of Fig. 2b). In addition, the corresponding toroidal H-field (lower left of Fig. 2b) and E-field (lower right of Fig. 2b) are observed as expected at the orthogonal planes, consistent with the electromagnetic induction theorem prediction. The above analysis implies that the proposed dielectric resonator supports independent coupling with the E-field and H-field of the incident THz waves through the ED and MD resonances operating at different frequencies, while demonstrating excellent isolation between them. Based on this advantage, with arrayed dielectric microspheres, if these two

resonances can be noninterferingly tuned by changing the microsphere spacing (e.g., once the ED resonance is tuned, the MD resonance always remains stable), a bidirectional strain sensor will be theoretically realized.

To verify this hypothesis, an array with an initial periodicity of $P_x = P_y = 300$ μm (Fig. 2c) is constructed with the predesigned microsphere. Then, a controlled-variable approach is adopted to reveal the influence of microsphere spacing ($P_x$ and $P_y$) on the ED and MD resonances. As shown in Fig. 2d, the increase in $P_x$ ($P_y$ is fixed) induces the ED resonance to move towards lower frequencies, but the MD resonance remains at approximately 0.3260 THz. For comparison, when $P_y$ increases from the preset value, the opposite phenomena are observed in Fig. 2e. The MD resonance shows a significant redshift, while the ED resonance remains at 0.4510 THz, which agrees well with our assumption (the corresponding simulated THz spectra are provided in Supplementary Fig. S1).

After completing this feasibility analysis, the physical principles behind these phenomena were further explored. Prior to that, the coupling types between the dipoles are defined as follows: side-by-side alignment of the dipole moments denotes transverse coupling, and the counterpart with end-to-end alignment is longitudinal coupling (inset of Fig. 2c). Then, we identified the physical mechanism of the

resonance change law considering dipole coupling theory[39–41]. In Fig. 2d, e, the E-field distribution (on the center x-y plane of the dielectric microsphere) for the ED resonance and the H-field distribution (on the center x-y plane of the dielectric microsphere) for the MD resonance are provided for comparing the mutual coupling varying between the initial state and the strain applied one. As shown in insets (i) of Fig. 2d, e, the moment of the single ED (namely, the local coupling intensity) remains stable regardless of the period increasing in the x- or y-direction. Therefore, the dominant factor contributing to the ED resonance frequency shifting can only be attributed to the mutual coupling format between the adjacent dipoles – i.e., transverse coupling presented in inset (i) of Fig. 2d rather than longitudinal coupling (inset (i) of Fig. 2e). Moreover, a similar conclusion that the MD resonance frequency shifting is determined by the transverse coupling between the MDs can be drawn by comparing insets (ii) of Fig. 2d, e. To further quantify the ED and MD resonance frequency shifting as a function of $P_x$ and $P_y$, respectively, a Lagrangian model is applied by solving the Euler-Lagrangian equations of motion to reveal the effect of the transverse coupling and longitudinal coupling as follows (see Supplementary Fig. S2 and Supplementary Note S2 for the detailed derivation process):

$$f_s^{ED} \approx f_0^{ED} \sqrt{1 + \kappa_{ET}^{ED} - \kappa_{EL}^{ED}} \qquad (1)$$

$$f_s^{MD} \approx f_0^{MD} \sqrt{\frac{1}{1 + \kappa_{HL}^{MD} - \kappa_{HT}^{MD}}} \qquad (2)$$

where $f_0^{ED}/f_0^{MD}$ and $f_s^{ED}/f_s^{MD}$ denote the initial frequencies and coupled frequencies of the ED/MD resonance. $\kappa_{ET}^{ED}/\kappa_{EL}^{ED}$ represents the transverse/longitudinal interaction coefficient between the EDs, while $\kappa_{HT}^{MD}/\kappa_{HL}^{MD}$ is the counterpart between the MDs. Because transverse coupling plays a dominant role in tuning the ED and MD resonances, $\kappa_{ET}^{ED}$ and $\kappa_{HT}^{HD}$ are the key parameters to be considered. From the aspect of the dipole interaction energy ($V_{ET}^{ED}$ and $V_{HT}^{MD}$), $\kappa_{ET}^{ED}$ and $\kappa_{HT}^{MD}$ can be linked to the periods $P_x$ and $P_y$ via:

$$\kappa_{ET}^{ED} \propto V_{ET}^{ED} = \frac{p_e^2}{4\pi\varepsilon_0 P_x^3} \qquad (3)$$

$$\kappa_{HT}^{MD} \propto V_{HT}^{MD} = \frac{p_h^2}{4\pi\varepsilon_0 P_y^3} \qquad (4)$$

where $p_e$ and $p_h$ are the magnitudes of the dipole moments of the ED and MD, respectively. $\varepsilon_0$ is the dielectric constant of free space. Based on the above analysis (Fig. 2d, e), the $p_e$ and $p_h$ amplitudes remain stable across varying excitations of the THz field. Therefore, according to Eq. (3), the dominant transverse ED coupling gradually decays as $P_x$ increases. Then, by substituting the reduced $\kappa_{ET}^{ED}$ into Eq. (1), a remarkable redshift of $f_s^{ED}$ can be derived, consistent with the previous trend shown in Fig. 2d. Continuing this analysis, the trend of $f_s^{MD}$, as shown in Fig. 2e, can also be explained in detail.

Thus, a theoretical model based on a dielectric microsphere array has been built to bridge the resonance shifting with the microsphere spacings along orthogonal (x- and y-) directions. When illuminated with y-polarized THz waves, the ED resonance is stimulated and moves towards a lower frequency as the x-directional microsphere spacing increases, while the MD resonance remains almost unchanged. The opposite phenomenon (the ED resonance remains stable, but the MD resonance shifts to a lower frequency) occurs for the MD resonance when $P_y$ is magnified, providing guidelines for designing an orthogonally bidirectional strain sensor.

## Meta-sensor array architecture and characterization

Based on the design strategy described above, a meta-sensor array consisting of a ZrO₂ microsphere array (array period $P_x = P_y = 320$ μm, microsphere diameter $d = 161$ μm) embedded in an elastomeric PDMS substrate (with a thickness of ~320 μm for efficiently transmitting THz waves shown in Supplementary Fig. S3) is designed (Fig. 3a) by analyzing the influence of the PDMS matrix on the inherent ED/MD resonance (Supplementary Fig. S4 and Supplementary Note S3), and fabricated (Fig. 3b and Supplementary Fig. S5) to independently detect and quantify the bidirectional strain. The relative permittivity of ZrO₂ and PDMS are 33 and 2.4, respectively. The relevant fabrication process is shared in Supplementary Note S4, and a simple micro template-assisted assembly strategy was adopted to array the ZrO₂ microspheres, by which the working area of our sensor array can be flexibly extended according to the easily fabricated template size. In turn, a sample with a size of $110 \times 130$ mm² (the largest one to our best knowledge) is implemented.

Prior to investigating the strain sensing behavior of the well-designed meta-sensor array, the mechanical properties of the fabricated sample and a pure PDMS film have been tested. Despite exhibiting a reduced stretchability compared to pure PDMS ($\varepsilon = 243\%$), our device can still withstand a mechanically fractured tensile strain of $\varepsilon = 156\%$ (Fig. 3c). Further, a THz time-domain spectroscopy is utilized to check the transmission spectral information when external single-directional stress (along the x- or y-direction) is applied to the sample. The corresponding results are shown in Fig. 3d, e. When the sample is stretched along the x-direction, the measured ED resonance gradually shifts from 0.4216 THz to 0.3507 THz (right side of Fig. 3d) as the strain increases from 0 to 65%, which is in accordance with the simulated results (left side of Fig. 3d). Note that when the applied strain exceeds 65%, the ED resonance will become unrecognizable due to the attenuation of the transmission amplitude (Fig. 3d). In contrast, as the y-directional strain increases from 0 to 70%, the measured MD resonance (left side of Fig. 3e) is the same as the simulation (right side of Fig. 3e) and gradually decreases to 0.3049 THz from 0.3218 THz. (see Supplementary Fig. S6 for analyzing the y-directional strain detection limit ~70%) It is worth noting that the remaining resonance (e.g., MD resonance during stretching in the x-direction, ED resonance during stretching in the y-direction) in these two processes barely shifts (Fig. 3f, g), further proving the dominant contribution of transverse coupling to resonance shifting. The Poisson's ratio of the substrate (0.4) was considered when conducting the simulations. These results indicate that this meta-sensor can simultaneously sense the strain amplitude and direction through the independent responses of ED and MD resonances to the strains in the x- and y-directions, respectively. This meta-sensor also exhibits excellent durability and strong stability. The transmission signal passing through the sample remains stable over 5000 tensile cycles, during which the applied strain is repeatedly varied from 0 to ~65% @ x direction/~70% @ y direction (Supplementary Fig. S7).

In addition, the sensitivity of our meta-sensor has also been analyzed from the aspect of the smallest detectable strain and the minimum detectable strain variation. Considering the spectral resolution of the THz time-domain spectroscopy system (0.001 THz), the theoretically smallest detectable strain values can be determined by shifting the resonance frequencies to lower frequencies by 0.001 THz from the initial 0%-strain states (0.4215 THz @ x-directional strain; 0.3220 THz @ y-directional strain). This results in a minimum detectable strain of 1.25% @ x-directional strain and 2.7% @ y-directional strain, as shown in insets (i) of Fig. 3f, g. Using the similar analysis approach, the theoretically minimum detectable strain variations of our meta-sensor also were obtained as 0.63% @ x directional strain (from 64.37% to 65%) and 2.7% @ y-directional strain (from 0 to 2.7%) (inset (ii) of Fig. 3f and inset (i) of Fig. 3g). (see Supplementary Note S5 for analyzing the minimum detectable strain variations) Then, the

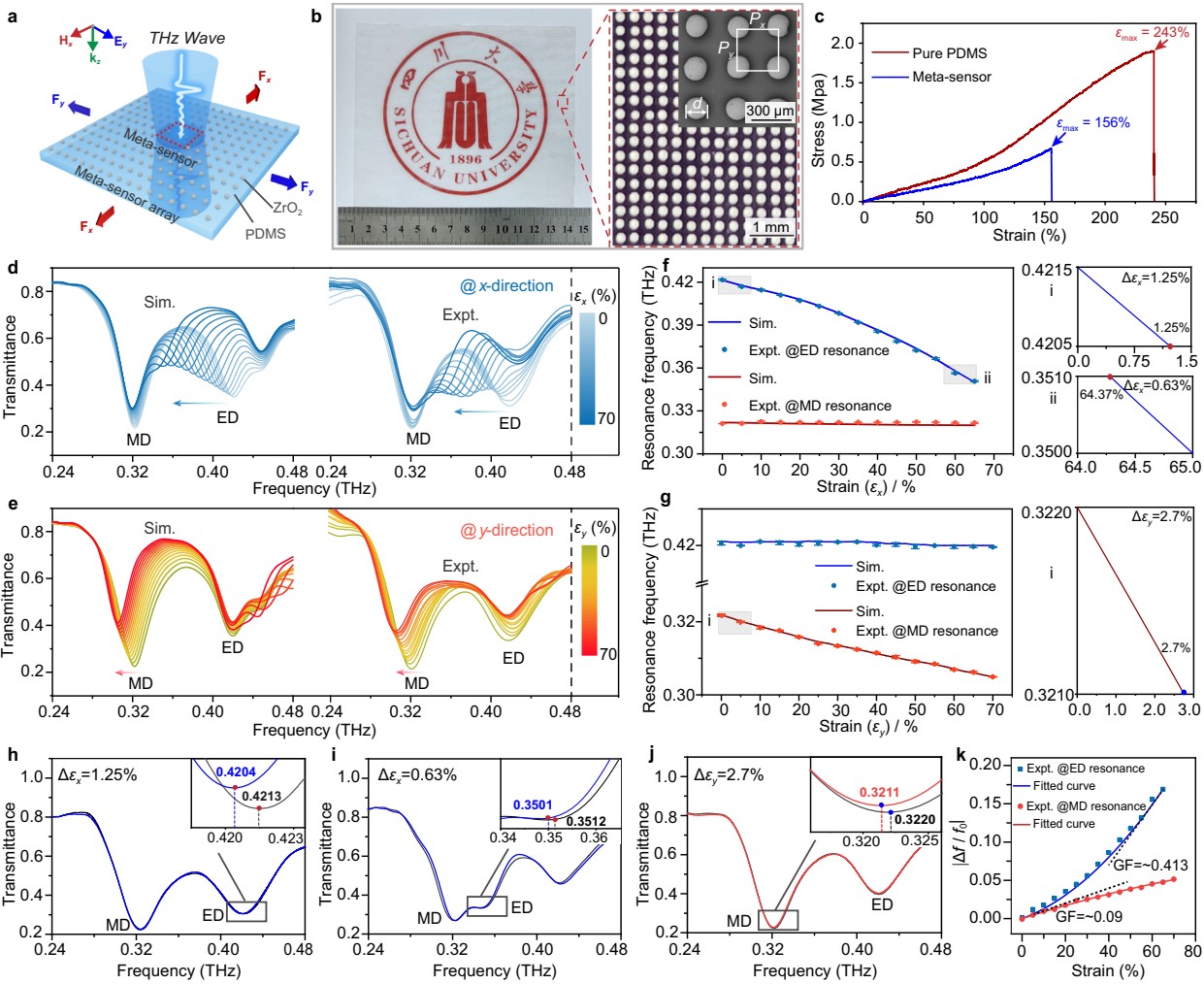

**Fig. 3 | Unidirectional tensile strain detection performance evaluation.**
**a** Schematic diagram of the proposed meta-sensor array with response to tensile force along the *x*- and *y*-axis directions. **b** Optical image of the meta-sensor array and magnified SEM images within the dashed square. **c** Measured stress-strain curves of a pure PDMS film and the fabricated meta-sensor array. Simulated and experimental transmittance spectra of the meta-sensor array applied with different tensile strains (varying from 0 to 70% by 5% per step): **d** for the *x*-direction; **e** for the *y*-direction. Extracted resonance information from **d** and **e** are plotted in **f** and **g**. The insets of **f** and **g** present the simulated smallest strains (i in **f** and **g**) and minimum strain variations (ii in **f** and i in **g**) that can be recognized. The data in **f** and **g** are presented as mean ± s.d. of *n* ≥ 6 independent measurements. **h**–**j** Corresponding measured results for verifying the smallest strains and minimum strain variations. **k** Relative resonance frequency variation (Δ*f*/*f*₀)-strain relationship.

corresponding experiments under the aforementioned strain states were conducted, and -0.001-THz resonance frequency shifting is observed (Fig. 3h–j and Supplementary Fig. S8), which is consistent well with the simulated predictions. Further, the maximum gauge factor (GF) of our device is also calculated (Supplementary Note S6), which is as -0.413 @ *x* directional strain and -0.09 @ *y* directional strain (Fig. 3k). While the GF and stretchability of our meta sensor may not outperform existing ultra-stretchable and highly sensitive strain sensors[42,43], our design excels in independently detecting bidirectional strain, recognizing arbitrary strain direction and mapping 2D strain distribution, which are still major challenges faced by current strain sensors.

Although our design has exhibited excellent performance for identifying preset strain along the *x*- or *y*-direction, the natural stress in real life is always along arbitrary directions. Therefore, based on the orthogonal decomposition feature, two orthogonally external forces (along the *x*- and *y*-directions) are simultaneously loaded at the proposed meta-sensor to evaluate its arbitrarily directional strain detection performance. As shown in Fig. 4a, with the *x*-directional strain varying from 4% to 32% by 4% per step, the strain applied in the *y*-direction is fixed at 4%, 8%, 12%, 16%, 20%, 24%, 28% and 32%. In such

cases, the ED resonance gradually shifts to a lower frequency as the *x*-directional strain increases, while the MD resonance always remains localized at a certain value only associated with the *y*-directional deformation ratio (4% @ 0.3213 THz, 8% @ 0.3199 THz, 12% @ 0.3187 THz, 16% @ 0.3178 THz, 20% @ 0.3167 THz, 24% @ 0.3160 THz, 28% @ 0.3154 THz, and 32% @ 0.3144 THz). By switching the strain loading, the opposite phenomenon can be revealed in Fig. 4b, i.e., the MD resonance frequency decreases with increasing strain in the *y*-direction, while the ED resonance always stays at a specific value, which is only related to the *x*-directional deformation ratio (4% @ 0.4177 THz, 8% @ 0.4157 THz, 12% @ 0.4131 THz, 16% @ 0.4104 THz, 20% @ 0.4059 THz, 24% @ 0.4000 THz, 28% @ 0.3946 THz, and 32% @ 0.3894 THz). These phenomena demonstrate that our design favors independent and noninterfering monitoring of the orthogonal strains. Additionally, the experimental results agree well with the simulated results shown in Supplementary Fig. S9, further proving the veracity of our strategy.

Furthermore, in Fig. 4c, d, the dynamic ED and MD resonance frequencies during stretching have been extracted from Fig. 4a, b to directly exhibit the corresponding relation between the bidirectional strains applied to our sensor and its resonance shifting. This facilitates

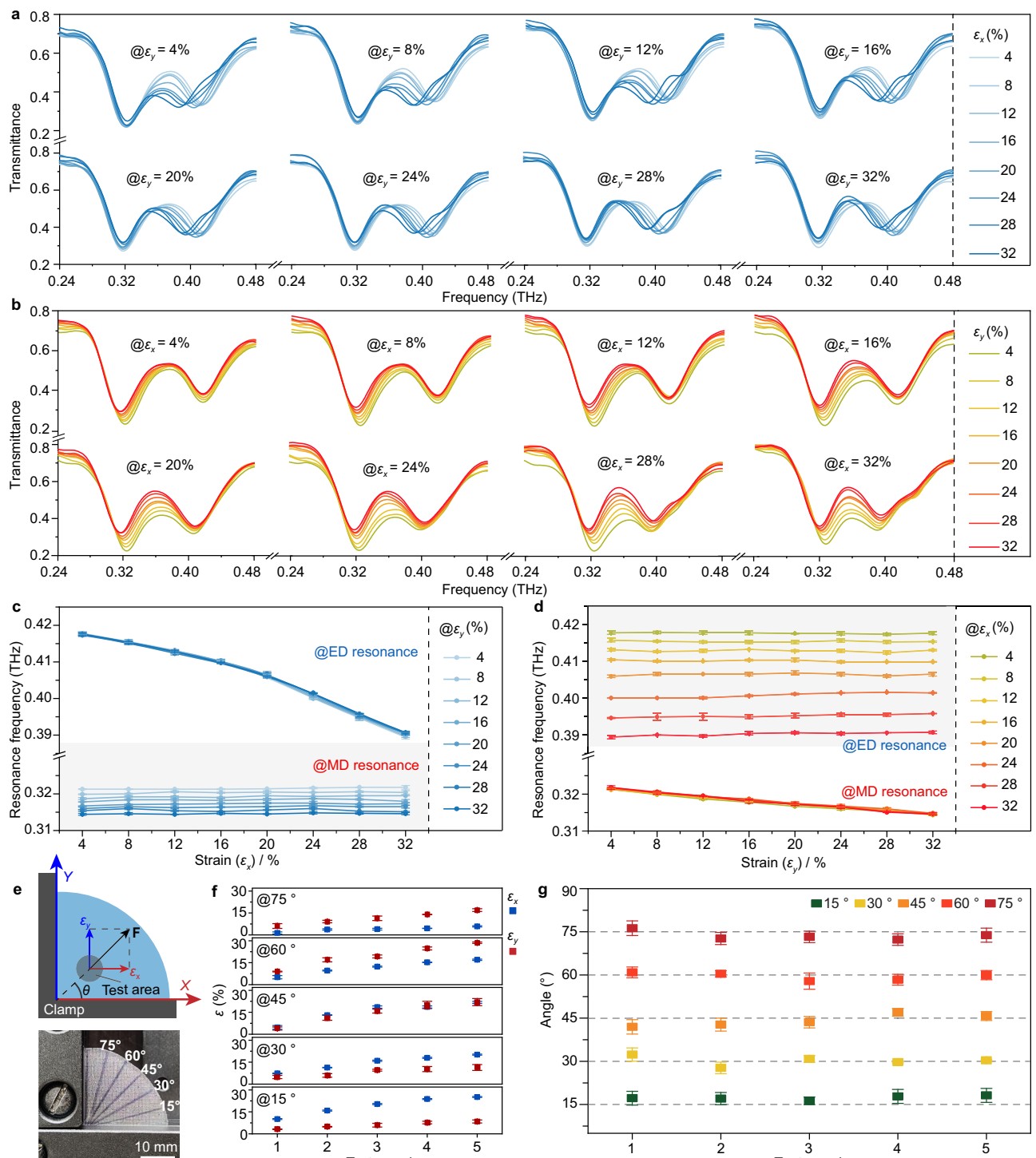

**Fig. 4 | Bidirectional tensile strain and strain direction detection performance evaluation.** Experimental transmittance spectra of the meta-sensor array simultaneously applied with $x$- and $y$-directional tensile strain: **a** $x$-directional tensile strain varying from 4% to 32% while the orthometric strain is maintained at 4%, 8%, 12%, 16%, 20%, 24%, 28%, and 32%; **b** $y$-directional tensile strain varying from 4% to 32% while the orthometric strain is maintained at 4%, 8%, 12%, 16%, 20%, 24%, 28% and 32%. Extracted resonance information from **a** and **b** are plotted in **c** and **d**. **e** Schematic and photograph of the setup and the sample for conducting the strain direction detection. **f** $\varepsilon_x$ and $\varepsilon_y$ measured by our device that is applied with the external strains along the direction of $\theta = 15°$, $30°$, $45°$, $60°$, and $75°$, respectively. **g** Calculated strain directions based on the measured $\varepsilon_x$ and $\varepsilon_y$. The data in **c**, **d**, **f**, and **g** are presented as mean ± s.d. of $n \geq 6$ independent measurements.

its practical applications in arbitrary-directional strain recognition. To further investigate the strain direction detection performance, a test setup was built, as depicted in Fig. 4e. In this setup, a right-angle clamp was adopted to securely hold fix the sample and effectively resist any correlative deformation induced by the Poisson's ratio of the PDMS matrix. In the meantime, the sample was reshaped into a quarter disk

with the radius of 3 cm (Fig. 4e), facilitating precise calibration of the stress direction. In the confirmatory experiments, the sample was stretched in 5 directions ($\theta = 15°$, $30°$, $45°$, $60°$, and $75°$) and in each direction five stretches with gradually increasing strain were conducted. During stretch, the corresponding strains along $x$ ($\varepsilon_x$) and $y$ ($\varepsilon_y$) directions were independently picked out by our meta-sensor array

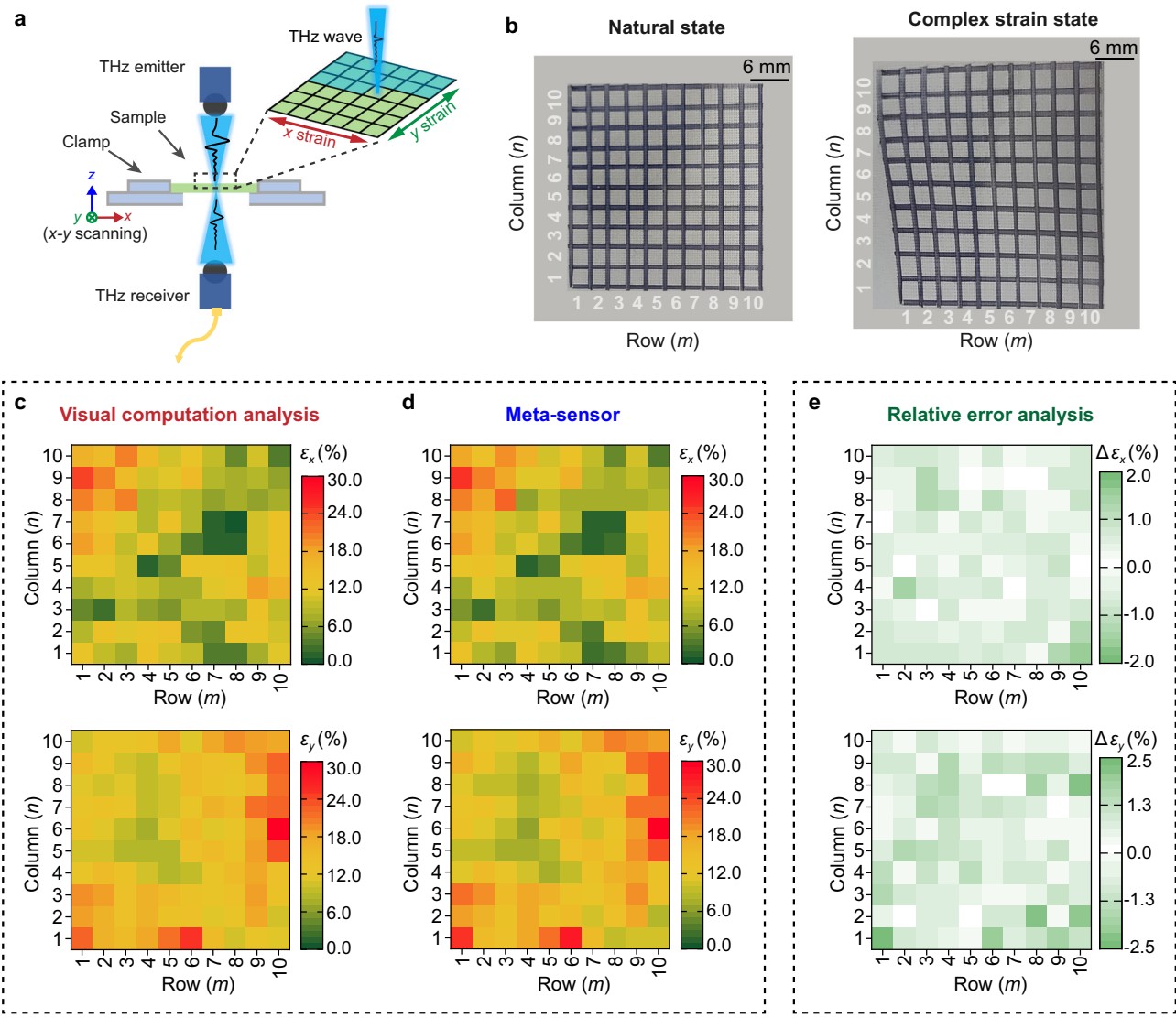

**Fig. 5 | Tensile strain mapping performance evaluation. a** Schematic of the 2D THz scanning platform for in-plane strain measurement. **b** Photographs of the sample in its natural state (left side) and under an arbitrary complex tensile strain (right side). A grid array is painted on the surface of the sample for the visual computation of the prestrain. Strain images (upper for *x*-directional deformation and lower for *y*-directional deformation) measured by visual computation analysis (**c**) and the meta-sensor array (**d**). **e** Relative test errors of our meta sensor array through referring to the visual computational results: upper for *x*-directional strain and lower for *y*-directional strain.

(Fig. 4f, and the corresponding experimental transmittance spectra are supplied in Supplementary Fig. S10). Eventually, following the principle of orthogonal decomposition, the strain directions can be accurately recognized and quantified by comparing the measured transverse and longitudinal strains using the equation: $\theta = \arctan(\varepsilon_y / \varepsilon_x)$. The minor discrepancies shown in Fig. 4g effectively confirm the strain direction detection ability of our meta sensor.

In addition to these advantages, our design undoubtedly offers the unique feature of mapping the in-plane strain distribution with high resolution due to its small meta-sensor and large density array. The size of a single meta-sensor (~3 × 3 mm², approximately four operating wavelengths) is determined by comprehensively considering the maximum detection resolution and minimum boundary impact. To prove this outstanding functionality, the sample subjected to a complex strain was tested with the help of a 2D THz scanning platform (Fig. 5a). This prestrain was generated by a two-axis stretching clamp (Supplementary Fig. S11). Before testing, the surface of the sample was marked with a 10 × 10 grid array shown in Fig. 5a, b (3 × 3 mm² per pixel is the same-sized as the single meta-sensor). The

state of the grids before and after stretching was recorded photographically (Fig. 5b), and then the real-applied strain was calculated through visual data-based computation (Fig. 5c). Simultaneously, the transmission spectrum information (Supplementary Table S2) of the sample was measured by the 2D scanning setup supporting the grid-by-grid measurement (Fig. 5a), for which the THz focal spot and its scanning step were both set to 3 mm in accordance with the size of the single meta-sensor and per grid. Then, according to the relationship between the strain magnitude and frequency shift established previously, the surface strain state of the sample is supplied in Fig. 5d, which is highly consistent with the visual data-based computation results (Fig. 5c, and the recorded visual data are supplied in Supplementary Tables S3 and 4). To further quantify the test error of our strategy, the relative error to the visual computational analysis (regarded as exact strain values) was calculated in Fig. 5e. The maximum test error occurs at (10, 1) (denoting the coordinates of the grid; the former for row number and the latter for column number) regarding the *x*-directional deformation (upper in Fig. 5e), and only reaches 1.8%. The test error for the *y*-directional deformation (lower in

Fig. 5e) is also smaller than 2.5% – the maximum relative error locating at grid (1, 1), and thus the impressive test accuracy endows our device great potential for real-life strain measurement.

In summary, we reported a meta-sensor array that can realize high-resolution identification and quantification of in-plane strain states over a large area ($110 \times 130$ mm$^2$). Our meta-sensor array comprises high-permittivity ZrO$_2$ microspheres in a square array encapsulated within a PDMS substrate through a micro template-assisted assembly strategy. Microsphere spacing in the $x$- and $y$-directions independently affect the ED and MD resonance frequencies, respectively. Considering their linear relationship when the external strain is no more than 28%, the strain information across our meta-sensor array can be quantified. Particularly, combined with the existing THz scanning technology, this meta-sensor array successfully mapped the direction and magnitude of an arbitrary strain distributed across a 2D plane with the relative test error superior to 1.8%. Overall, our meta-sensor array offers several attractive features compared with previous strain sensor arrays, such as the capability to recognize strain direction, high resolution, large area coverage and low cost (Supplementary Table S1). Furthermore, the fabrication process is compatible with other surface treatment technologies, such as superhydrophobic treatment, endowing the meta-sensor with a self-cleaning capacity to avoid environmental interferences (such as dust and water) (Supplementary Fig. S12, Supplementary Note S7 and Supplementary Video S1–4). Although the transmission-type THz meta sensors have been developed for the detection of strain magnitude and directions, it is indeed faced with the challenge that THz signal can hardly penetrate human body. With the strategy we proposed, a reflection-type meta-strain-sensor was achieved to address this issue, and the feasibility has been verified by simulation and experiment (Supplementary Note S8 and Supplementary Figs. S13–15).

## Methods

### Fabrication of strain meta-sensor array

The fabrication process of the strain meta-sensor array is shown in Supplementary Fig. S5. The ZrO$_2$ microspheres of suitable and homogeneous size were first selected based on simulation analysis. Subsequently, to obtain an even and flat supporting layer for ensuring the microspheres distributed on the same horizontal plane in the PDMS matrix, liquid PDMS (the base and crosslinking agent at a weight ratio of 10:1) after defoaming was scraped onto a polyethylene terephthalate (PET) supporting substrate by a high precision scraper (HQ-TB-G, Huaqi Instrument Co., LTD, China) with the accuracy of ±0.5 μm. Then, it was placed in an oven at 70 °C for ~10 min to pre-cure the PDMS adhesive layer (~80 μm). Particularly, a PET supporting substrate is necessary for the fabrication process of the sample to avoid curling of the ultra-thin soft PDMS film.

Next, the screen-printing template (mesh slightly larger than the diameter of the microsphere) was attached to the PDMS adhesive layer and the selected microspheres were placed on it. Then, the microspheres were pushed into the holes and assembled as an specific array with the help of a soft brush providing an external force ($F_e$) shown in Supplementary Fig. S5a. After removing the template, the above structure was encapsulated with liquid PDMS and further cured in the oven at 70 °C for ~25 min. Finally, the predesinged stretchable meta-sensor array is achieved by peeling it from the PET supporting substrate. With our method, the ZrO$_2$ microspheres are nearly on the same horizontal plane (Supplementary Fig. S5b) and the microsphere position deviation has little effect on the overall performance of our device (Supplementary Fig. S16).

### THz time-domain spectroscopy measurement

The THz time-domain spectroscopy (QT-TRS1000, Quenda, China) was used to measure the transmittance of the meta-sensor sample at various strain states. First, the sample was fixed by the self-made stretching fixtures (Supplementary Figs. S11 and S17), by which an external strain can be flexibly loaded as needed. The incident THz wave was then collimated and focused on the sample, and the THz receiver collected the time-domain signals with sample information. After Fourier transform of time-domain signals (meta-sensor sample) and normalization with the reference signal (without sample), the transmittance spectra were obtained. When conduct the measurement, each THz signal was collected 100 times per test round for calculating the average value as well as six sets of parallel tests for determining the corresponding standard error.

## Data availability

The data supporting the findings of this study are available within the Article and its Supplementary Information. Source data are provided with this paper. Other raw data generated during this study are available from the corresponding authors upon request. Source data are provided with this paper.

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

## Acknowledgements

This study was financially supported by the National Key Research and Development Program (2023YFB3811300(W.H.), 2023YFB3811303(W.H.)), the National Science Foundation (NSFC) for Distinguished Young Scholars of China (62225108(Q.C.)), the National Natural Science Foundation of China (U20A20212(W.H.)), the Program of Song Shan Laboratory (Included in the management of Major Science and Technology Program of Henan Province) (221100211300-02(Q.C.)), the National Natural Science Foundation of China (62288101(Q.C.), 61731010(Q.C.), 62101394 (C.Z.)).

## Author contributions

X.L., F.Z., C.Z., W.H. and Q.C. conceived and designed the research. X.L., F.Z. and W.H. designed, fabricated and characterized the devices. X.L. performed the electromagnetic simulations under the supervision of Q.C. and C.Z. F.Z. helped superhydrophobic testing. X.L., C.Z. and Q.C. contributed to the data analysis and mathematical analysis. L.Z., J.Y., S.P., X.C., H.Z., K.C. and Q.S. assisted with the device fabrication and measurement. X.L, F.Z. and C.Z. wrote the manuscript. All the authors discussed and commented on the manuscript.

## Competing interests

The authors declare no competing interests.
