## [Peer Review File · Nature Communications]

A terahertz meta-sensor array for 2D strain mappingREVIEWER COMMENTS

Reviewer #1 (Remarks to the Author):

The manuscript entitled “A terahertz meta-sensor array for 2D strain mapping” mainly reports a meta-sensor array composed of dielectric resonators and flexible PDMS substrate. When the strain is applied to the device, the mode profile of the dielectric resonators will be changed, resulting in frequency and amplitude differences of both MD and ED modes. Based on this principle, the authors proposed strain mapping application by making the devices into an array.

Overall, I don't find too much novelty of this work. Many related stretchable metamaterials have been reported before. Even in terahertz region, there have been plenty of works (Advanced Optical Materials, 2019, 7(19): 1900379; Optics letters, 2013, 38(12): 2104-2106), but the authors didn't mention these in their manuscript and make any comparisons regarding the performances. Although the authors leveraged the dielectric resonators rather than the plasmonic ones, but they still didn't clearly explain the advantages in terms of performance and fabrication process. Moreover, the proposed application for strain mapping has also been reported recently at the same frequency band using THz metamaterials (APL Photonics, 2021, 6(11): 116105). Therefore, I don't consider for further consideration of publication at Nature Communications.

Besides, I also have several technical concerns:

1. Many results lack statistic analysis. What is the reproducibility of each result?
2. The resonant frequencies of MD and ED seems to be fixed. Are there any design methodologies for independent control of each mode?
3. What is the minimum strain that can be detected by the device, and what is the maximum strain can be applied to the device?

Reviewer #2 (Remarks to the Author):

The authors novelty applied Mie resonance to strain measurement and realized in plane strain recognition. But the authors should complement the details of the strain sensing and the applications. This manuscript could be accepted after fulfilling the experiments and modifications mentioned below.

1. The data shown in figure 3e and 3f, figure 4c and 4d was not enough, each line in those figures only possessed five points, which is not sufficient. The authors should measure the resonance frequency under more different strains. And the authors should measure the strain detection limit and the smallest strain variation that could change the resonance frequency.

2. As the authors mentioned in the abstract that arbitrary strain information could be detected. But the authors did not perform any experiments on the strain direction detection. The authors should design and perform more experiments to complement and support their conclusion.

3. Normally, mixing particles like ZrO₂ into PDMS would cause a significant decrease in mechanical properties. The authors should complement the experiments about mechanical properties.

4. The authors declared that a sensor array containing ~1580 meta sensors was prepared, but in the manuscript, the authors did not exhibit any sensor array with 1580 meta sensors. The authors only performed the application of the 6×6 sensor array. If this work could reach such a large number of sensors, the authors should design and perform more appropriate applications of the large sensor array.

5. In Figure 1 and Figure 3, the diagram shows that the structure of the sensor was the ZrO₂ particles uniformly dispersed in the PDMS, but in Figure 5 the photograph shows that the 6×6 sensor array presents a mesh-like structure. Is the sensor structure changed in the sensor array?

Reviewer #3 (Remarks to the Author):

The manuscript “A terahertz meta-sensor array for 2D strain mapping” by X. Lu et al. proposes a flexible meta-sensor array that can detect the in-plane direction and magnitude of preloaded strains by referencing a dynamically transmitted terahertz (THz) signal. Although terahertz metamaterial detection technology based on Mie resonances is not new, this is an interesting attempt on a topic of current interest. Some major concerns should be addressed to make the presentation more informative.

1. The idea of using dielectric particles to generate electric and magnetic resonances were previously reported ([https://doi.org/10.1016/S1369-7021\(09\)70318-9](https://doi.org/10.1016/S1369-7021(09)70318-9); <https://doi.org/10.1007/s00339-011-6716-2>). An efficient absorber in the THz band with a monolayer of zirconium dioxide (ZrO₂) microspheres was also demonstrated (<https://doi.org/10.1364/OE.26.013001>). But the authors haven't cited these references in the manuscript.

2. It is preferred to measure the strain sensor for wearable devices with a reflection THz system instead of a transmission system, as THz signal cannot penetrate human body, so the potential of this strain sensor for real applications is questionable.

3. In page 9, the authors described a strategy for fabrication of the metasensor array, noticed the diameter of ZrO₂ microsphere is 161 μm and the thickness of PDMS substrate is 320 μm, please clarify how to ensure that the microballs were distributed on the same horizontal plane embedded in the substrate. How to avoid frequency shift caused by different positions of microballs?

4. In Fig.3c, please describe why does the tensile force along x-direction not affect the frequency shift of the peak between ED and MD resonances, but it is observed in the simulation results? And please clarify why the parameter of the tensile strain is only set to 28%. What would happen if it is set higher?

5. Please provide additional data for y-directional tensile strain with 28% in Fig 4a and x-directional tensile strain with 28% in Fig 4b, which can make the results more convincing. And according to the author's description, are these data derived from an average of 5000 times? Please add scale bar in Fig. 5. Similarly, please provide the relevant data when the stress level reaches 28% according to the authors strategy.

6. The key parameters of a strain sensor include high stretchability and high sensitivity (gauge factor). In the manuscript, the measured strain step is 7% (Fig. 3), which actually is too large for a strain sensor. The authors should calculate the gauge factor of their design. The stretchability of this

manuscript is below 30%, which is lower than the reported strain sensor using the same material-PDMS (<https://doi.org/10.1021/nn501204t>). An ultra-stretchable and highly sensitive strain sensor has been reported with an ultra-high stretchability of 1000% and high sensitivity with a maximum gauge factor (GF) of up to 165 (DOI <https://doi.org/10.1039/C8NR08589G>). The authors should make a comparison with these results.

Reviewer #4 (Remarks to the Author):

In this manuscript, the authors proposed a simple and unique meta-sensor array that can detect the strains by referring to the dynamically transmitted terahertz (THz) signal. The strain sensing mechanism based on Mie resonance is discussed in detail and the correspondence between the electric/magnetic dipole resonance frequency and the horizontal/vertical tension level is established. The information on strain magnitude and direction at any position in the plane can be obtained through the proposed strategy, as demonstrated by the experimental results. In addition, they also demonstrate the unique large-area preparation process and self-cleaning function of meta-sensor array.

Overall, the paper is very nicely written and well structured. Considering the significance and broad-ranging applications of the developed strategy, I would recommend its publication in Nature Communications, after the following minor issues are addressed.

(1) In the experiment, the zirconia (ZrO₂) microsphere array in a polydimethylsiloxane (PDMS) substrate was utilized to prove the proposed strain sensing mechanism, while in the "Bidirectional strain detection strategy" part, the sensing mechanism is established in a vacuum background. Thus, the authors should discuss the effect of PDMS on electric/magnetic dipole resonance and the spectral information of pure PDMS in the THz band should be supplemented.

(2) The meta-sensor array proposed in the manuscript has a sensor density of ~11.1 cm⁻², please explain the origin of this value. Since the proposed meta-sensor array is not the same as the conventional electrical sensor array, how do the authors define the "sensing density" of this meta-sensor array?

(3) In Supplementary Fig. S4, SEM image of the section of the meta-sensor array is presented. However, the microsphere arrays are not positioned in the middle layer of the encapsulation layer. The reviewer would like to know if the thickness of the PDMS encapsulation layer film has any effect on the device performance?

(4) In Supplementary Fig. S6, please provide more information on the cyclic stability during strain, not just after the release of the stretch cycle.

(5) Do superhydrophobic micro-nano structures have an effect on resonance signals? Please provide relevant explanations.

(6) In Video S3, Video S4, please add some captions to make it easier for the reader to understand.

**Point-by-point response to reviewer:**

**Response to Reviewer #1**

Reviewer #1 (Remarks to the Author):

*The manuscript entitled “A terahertz meta-sensor array for 2D strain mapping” mainly*
*reports a meta-sensor array composed of dielectric resonators and flexible PDMS*
*substrate. When the strain is applied to the device, the mode profile of the dielectric*
*resonators will be changed, resulting in frequency and amplitude differences of both*
*MD and ED modes. Based on this principle, the authors proposed strain mapping*
*application by making the devices into an array.*

*Overall, I don't find too much novelty of this work. Many related stretchable*
*metamaterials have been reported before. Even in terahertz region, there have been*
*plenty of works (Advanced Optical Materials, 2019, 7(19): 1900379; Optics letters,*
*2013, 38(12): 2104-2106), but the authors didn't mention these in their manuscript and*
*make any comparisons regarding the performances. Although the authors leveraged*
*the dielectric resonators rather than the plasmonic ones, but they still didn't clearly*
*explain the advantages in terms of performance and fabrication process. Moreover, the*
*proposed application for strain mapping has also been reported recently at the same*
*frequency band using THz metamaterials (APL Photonics, 2021, 6(11): 116105).*
*Therefore, I don't consider for further consideration of publication at Nature*
*Communications.*

**Response:** Thank you very much for your valuable feedback, and we apologize for
missing certain pertinent comparisons with stretchable metamaterials in the previous
manuscript. Accordingly, we have added the introduction of the reviewer-mentioned
works in this revised manuscript (lines 79-88, page 3) as:

“On the other hand, stretchable metamaterials, as an alternative strategy, possess
inherent advantage in constructing continuously large-scale sensor array and sensors
based on polarization-dependent resonance-strain response have been achieved by
patterning metal films on elastomer substrates³¹⁻³³. However, limited by the inevitable
correlative deformations in metallic patterns (coupled x- and y-directional

deformations), they are only equipped with small ultimate-strain sensing capability (up
to 6%) and powerless in strain direction recognition. In addition, due to the limited
interface bonding and mismatched mechanical properties, metallic patterns in existing
plasmonic metamaterials tend to be damaged or even peel off the substrate after massive
cyclic stretching, thereby constraining their overall performance and durability.”

Although the plasmonic metamaterials reported in your mentioned works [1-3] can
assess external strain by observing the resonance frequency shifts, it’s important to note
that their operational principles are entirely distinct from ours. Furthermore, their
performance in strain detection significantly lags behind ours, particularly when it
comes to detecting strain directions. Regarding the plasmonic ones in Refs. [1-3], the
resonance shifting is induced by the capacitance change directly related to the spacing
of meta-units, following the principles of a classical RLC circuit. Therefore, the
corresponding resonance-strain response is polarization-dependent, namely the
resonance shifting can only be aroused when the strain is applied along the polarization
direction of the incident waves. In contrast, our design can independently detect the
bidirectional strain by monitoring the frequency shifts of the inherent ED and MD
resonances, which are simultaneously induced by dielectric particles interacting with
electromagnetic waves and independently tuned by changing their orthogonally
transversal couplings directly related to the corresponding directional spacing of meta-
units (for a detailed explanation of the working principle, please refer to the
“Bidirectional strain detection strategy” section in the main text and Note S2 in the
Supporting Information). Based on this unique feature, our device integrates
comprehensive strain detection capabilities, including independent bidirectional strain
detection, 2D strain mapping, and strain direction recognition, marking a pioneering
achievement in this field.

Then to highlight our novelty, we have carefully compared our work with those in Refs.
[1-3] in terms of performance and fabrication process:

**1. Working performance**

Although those works in Figs. R1-R3 can judge the external strain though referring to
 the resonance frequency shifting, our design shows the unique working performance
 than others in Refs. [1-3].

1.1 Polarization-independent detection strategy

In Refs. [1-3], their resonant behaviors of the strain-loaded metamaterials are all
 polarization-dependent in a manner that can only be caused by a strain along the
 polarization direction of the incident waves (Figs. R1-R3). Consequently, if the
 direction of the external strain is unknown beforehand, these devices cannot directly
 measure the applied strains. Instead, a complex two-time test involving quadrature
 polarized incident field must be conducted first to determine the strain direction. In
 contrast, our strategy enables the simultaneous detection of the applied transverse and
 longitudinal strains in a single step by observing the resonance shifts of ED and MD
 without changing the polarization state of the incident THz wave (as demonstrated in
 Figs. 2-5 of the revised manuscript). This simplifies the testing process and enhances
 its practical application.

 Fig. R1. (a) Schematic illustrations of the proposed PSM device (reported in Ref. [1]) operated by
 applying a stretching force along the x - and y -axis directions. Experimental results and optical
 microscope images of a PSM device stretched with different w (b) and l (c) values in the TE and
 TM modes. Inserted scale bars are $50 \mu\text{m}$.

Fig. R2. (a) and (b) Schematics of the unit cell. (c) and (d) Numerical results for the transmission
 spectra with different gap widths, for the horizontal and vertical polarizations, respectively, when
 the strain is loaded along the x axis. The percentage values are the calculated strain of the unit cell.

Fig. R3. (a) Schematic of the passive strain sensor. (b) Simulated strain- and polarization-dependent
 terahertz transmission spectra, and s - to p -polarized terahertz transmission spectra of the MM
 composite for different levels of local strain. (c) Two-dimensional transmission ratio maps for a
 measured global strain of (a) 0%, (b) 2%, (c) 4%, and (d) 6%, with a scale bar in (a) of 20 mm.

1.2 Independently bidirectional-strain detection

Furthermore, for a high-quality bidirectional strain sensor, if strains in the x or y
 directions are fixed, the corresponding measurement should remain stable even when
 strains in the orthogonal direction vary arbitrarily. However, in the case of the metallic
 patterns shown in Fig. R1(a), the applied external strain along x/y direction induces
 their correlative deformations of metallic patterns in y/x direction. This results in a
 noticeable shift of the resonance associated with the non-stretching direction (as
 indicated by the blue line in Fig. R1(b) and orange line in Fig. R1(c)). Undoubtedly,
 this limitation poses a significant obstacle to achieving precise bidirectional strain

detection, especially under large strains. Naturally, this issue becomes less pronounced
 when the applied strain is maintained at a lower level, as illustrated in Figs. R2(c-d).
 Thus, we believe this is the reason why the strain mapping in Fig. R3(c) just can only
 be performed when the tension ratio is just below 6%. In contrast, our design can
 independently measure the x - and y -directional strain without mutually coupling (as
 demonstrated in Fig. 4 of the revised manuscript). This advantageous feature allows for
 accurate 2D strain mapping with our device (shown in Fig. 5 of the revised manuscript),
 even when the applied uniaxial tensile strain exceeds 60% (shown in Fig. 3 of the
 revised manuscript) or bidirectional simultaneous tensile strain exceeds 30% (shown in
 Fig. 4 of the revised manuscript), which is significantly greater than the range covered
 in Ref. [3].

1.3 Strain direction detection

Comprehensively considering these two aforementioned features, our design also
 provides a very meaningful application – recognize the direction of external strain
 based on the orthogonal decomposition principle. As shown in Fig. R4, the strain along
 15° , 30° , 45° , 60° , and 75° can be accurately recognized through comparing the
 measured transverse and longitudinal strains ($\theta = \arctan(\varepsilon_y/\varepsilon_x)$, where ε_x and ε_y
 denote the transverse and longitudinal strains, respectively). This achievement
 addresses a persistent challenge in current research. The corresponding description has
 been provided in lines 304-317, pages 13-14 as “...To further investigate the strain
 direction...” and new Figs. 4(e-g) in the revised manuscript.

Fig. R4. (a) Schematic and photograph of the setup and the sample for conducting the strain direction

detection. (b) ε_x and ε_y measured by our device that is applied with the external strains along the
 direction of $\theta = 15^\circ, 30^\circ, 45^\circ, 60^\circ,$ and $75^\circ,$ respectively. (c) Calculated strain directions based on
 the measured ε_x and $\varepsilon_y.$

2. Fabrication method

Different from the lithography processes in the aforementioned literatures, we propose
 a straightforward and low-cost micro-template-assisted assembly method. This
 approach enables a large stretchability of the devices and streamlines the fabrication of
 large-area samples, with a record-breaking size of approximately $110 \text{ mm} \times 130 \text{ mm}.$
 In the case of stretchable metamaterial sensors based on the plasmonic excitation
 principle, limitations arise from the bonding at the interface and disparities in expansion
 coefficients between the metallic patterns and the polymer substrates. Consequently,
 the fabricated patterns are prone to detachment or damage following repeated
 deformations, leading to the risk of performance degradation. In contrast, our device
 shows almost no performance degradation even after 5000 cycles of cyclic stretching
 (Fig. R5), exhibiting excellent stretchability and durability. Fig. R5 has been added in
 the Supporting Information as Fig. S7.

 Fig. R5. Measured ED and MD resonance frequencies of the fabricated sample after 1, 10, 50, 100,
 1000, 2000 and 5000 cycles of stretching (the applied strain repeatedly varies from 0 to $\sim 65\%$ @ x
 direction and from 0 to $\sim 70\%$ @ y direction): (a) for initial state (0% strain); (b) for $\sim 65\%$ strain
 along x direction; for $\sim 70\%$ strain along y direction.

 In conclusion, our meta-sensor operates on a fundamentally different principle
 compared to the plasmonic metamaterials. By adjusting the coupling of Mie resonance,
 outstanding abilities for accurately determining strain direction and performing 2D
 strain mapping can be achieved within a single device. In addition, a simple fabrication
 process for the meta-sensor has been proposed, that is, the microtemplate-assisted

assembly of ZrO₂ microsphere arrays followed by PDMS encapsulation. This process
opens up the possibility to achieve rapid and large-scale fabrication. The demonstrated
stretchable strain meta-sensor possesses high flexibility, applicability, and cost-
effectiveness for widespread THz device applications.

**References**

- 1. Xu Z, Lin Y S. A stretchable terahertz parabolic-shaped metamaterial[J]. *Advanced Optical*
*Materials*, 2019, 7(19): 1900379.
- 2. Li J, Shah C M, Withayachumnankul W, et al. Flexible terahertz metamaterials for dual-axis
strain sensing[J]. *Optics letters*, 2013, 38(12): 2104-2106.
- 3. Khatib O, Tyler T, Padilla W J, et al. Mapping active strain using terahertz metamaterial
laminates[J]. *APL Photonics*, 2021, 6(11).

*Besides, I also have several technical concerns:*

*1. Many results lack statistic analysis. What is the reproducibility of each result?*

**Response:** Thank for your valuable comments. Actually the experimental data in our
manuscript are presented as mean \pm standard deviation with of $n \geq 6$ independent
measurements, where the standard deviation is quite small.

In detail, to ensure the reliability of each result, each THz signal was collected 100
170 times per test round to calculate the average value, and we conducted six sets of parallel
tests for determining the standard deviation. Due to our negligence, it has not been
clarified in the previous manuscript. As your kind reminder, we have added the
corresponding description “...When conduct the measurement, each THz signal was
collected 100 times per test round for calculating the average value as well as six sets
of parallel tests for determining the corresponding standard deviation.” in the Methods
section (lines 406-408, page 17) of the revised manuscript.

Furthermore, to address your concerns, the data of the parallel experiments about the
unidirectional tensile strain detection (Figs. 3(d-g) of the revised paper) are provided in
Tables R1-4, where the average values and the standard deviation were also exhibited.

The small standard deviations (namely the error bars shown in Figs. 3(f-g)) affirm the
 excellent reproducibility of our meta-sensor for strain detection.

Table R1. MD resonance frequency statistics under strain in x direction

		ε_x @MD resonance							
Strain (%)	Samples	1	2	3	4	5	6	mean value	standard deviation
	0		0.3210	0.3213	0.3212	0.3210	0.3212	0.3212	0.3211
5		0.3210	0.3213	0.3212	0.3210	0.3214	0.3213	0.3212	0.0002
10		0.3230	0.3221	0.3225	0.3230	0.3222	0.3226	0.3226	0.0004
15		0.3220	0.3221	0.3220	0.3221	0.3222	0.3222	0.3221	0.0001
20		0.3220	0.3221	0.3220	0.3219	0.3221	0.3219	0.3220	0.0001
25		0.3220	0.3221	0.3220	0.3221	0.3221	0.3220	0.3221	0.0000
30		0.3220	0.3221	0.3220	0.3218	0.3220	0.3221	0.3220	0.0001
35		0.3220	0.3221	0.3220	0.3222	0.3219	0.3222	0.3221	0.0001
40		0.3220	0.3221	0.3220	0.3221	0.3222	0.3221	0.3221	0.0001
45		0.3220	0.3221	0.3220	0.3219	0.3221	0.3219	0.3220	0.0001
50		0.3220	0.3225	0.3221	0.3220	0.3227	0.3221	0.3222	0.0003
55		0.3220	0.3221	0.3220	0.3221	0.3219	0.3220	0.3220	0.0001
60		0.3220	0.3213	0.3217	0.3222	0.3212	0.3217	0.3217	0.0004
65		0.3220	0.3219	0.3210	0.3220	0.3219	0.3210	0.3216	0.0005

Table R2. ED resonance frequency statistics under strain in x direction

		ε_x @ED resonance							
Strain	Samples	1	2	3	4	5	6	mean value	standard deviation
	0		0.4220	0.4210	0.4220	0.4219	0.4209	0.4221	0.4216
5		0.4175	0.4168	0.4167	0.4173	0.4166	0.4166	0.4169	0.0004
10		0.4140	0.4151	0.4144	0.4140	0.4152	0.4145	0.4145	0.0005
15		0.4110	0.4112	0.4108	0.4108	0.4110	0.4107	0.4109	0.0002
20		0.4070	0.4070	0.4070	0.4073	0.4068	0.4072	0.4071	0.0002
25		0.4030	0.4030	0.4030	0.4031	0.4031	0.4033	0.4031	0.0001
30		0.3980	0.3985	0.3983	0.3982	0.3983	0.3983	0.3983	0.0001
35		0.3920	0.3919	0.3920	0.3919	0.3917	0.3921	0.3919	0.0001
40		0.3850	0.3861	0.3857	0.3848	0.3863	0.3857	0.3856	0.0006
45		0.37918	0.378	0.37859	0.3792	0.3778	0.3788	0.3786	0.0006
50		0.3720	0.3720	0.3730	0.3718	0.3721	0.3729	0.3723	0.0005
55		0.3660	0.3674	0.3660	0.3663	0.3673	0.3662	0.3665	0.0006
60		0.3560	0.3570	0.3560	0.3560	0.3570	0.3562	0.3564	0.0005
65		0.3500	0.3510	0.3510	0.3502	0.3512	0.3511	0.3507	0.0005

Table R3. MD resonance frequency statistics under strain in y direction

		ε_y @MD resonance							
Strain (%)	Samples	1	2	3	4	5	6	mean value	standard deviation
	0		0.3220	0.3219	0.3215	0.3218	0.3218	0.3212	0.3217
5		0.3200	0.3200	0.3200	0.3197	0.3202	0.3198	0.3200	0.0002
10		0.3180	0.3187	0.3184	0.3181	0.3189	0.3183	0.3184	0.0004
15		0.3175	0.3176	0.3176	0.3177	0.3174	0.3178	0.3176	0.0001
20		0.3160	0.3156	0.3158	0.3159	0.3157	0.3159	0.3158	0.0002
25		0.3142	0.3140	0.3141	0.3141	0.3137	0.3140	0.3140	0.0002
30		0.3135	0.3133	0.3134	0.3137	0.3134	0.3131	0.3134	0.0002
35		0.3127	0.3123	0.3125	0.3127	0.3123	0.3123	0.3125	0.0002
40		0.3115	0.3113	0.3114	0.3118	0.3115	0.3113	0.3115	0.0002
45		0.3100	0.3109	0.3105	0.3099	0.3107	0.3102	0.3104	0.0004
50		0.3094	0.3087	0.3091	0.3096	0.3088	0.3091	0.3091	0.0004
55		0.3084	0.3085	0.3084	0.3082	0.3088	0.3084	0.3084	0.0002
60		0.3070	0.3071	0.3070	0.3068	0.3070	0.3067	0.3069	0.0002
65		0.3060	0.3064	0.3062	0.3061	0.3065	0.3060	0.3062	0.0002
70		0.3050	0.3050	0.3050	0.3049	0.3048	0.3048	0.3049	0.0001

Table R4. ED resonance frequency statistics under strain in y direction

		ε_y @ED resonance							
Strain (%)	Samples	1	2	3	4	5	6	mean value	standard deviation
	0		0.4200	0.4210	0.4210	0.4199	0.4209	0.4210	0.4206
5		0.4200	0.4200	0.4200	0.4198	0.4200	0.4200	0.4200	0.0001
10		0.42104	0.421	0.421	0.4213	0.4209	0.4212	0.4211	0.0001
15		0.4210	0.4210	0.4200	0.4210	0.4212	0.4202	0.4207	0.0005
20		0.4210	0.4200	0.4200	0.4207	0.4199	0.4201	0.4203	0.0005
25		0.4210	0.4200	0.4210	0.4213	0.4198	0.4209	0.4207	0.0006
30		0.4210	0.4210	0.4210	0.4207	0.4210	0.4208	0.4209	0.0001
35		0.4210	0.4210	0.4210	0.4210	0.4212	0.4207	0.4210	0.0002
40		0.4200	0.4205	0.4200	0.4198	0.4204	0.4199	0.4201	0.0003
45		0.4200	0.4208	0.4200	0.4200	0.4210	0.4198	0.4203	0.0005
50		0.4200	0.4200	0.4190	0.4202	0.4200	0.4189	0.4197	0.0006
55		0.4199	0.4200	0.4200	0.4199	0.4200	0.4199	0.4199	0.0001
60		0.4199	0.4200	0.4190	0.4197	0.4202	0.4192	0.4197	0.0005
65		0.4196	0.4195	0.4198	0.4198	0.4194	0.4201	0.4197	0.0003
70		0.4195	0.4194	0.4195	0.4198	0.4195	0.4196	0.4196	0.0001

2. The resonant frequencies of MD and ED seems to be fixed. Are there any design
methodologies for independent control of each mode?

**Response:** We gratefully appreciate your valuable comment. Classical Mie resonance

theory indicates that the dielectric constant and diameter of the dielectric microspheres
determine the peak positions of the Mie resonance (including 1st-order MD resonance
and 1st-order ED resonance). Therefore, once the electromagnetic and structural
parameters have been established, the resonant frequencies of the MD and ED will be
fixed just as you thought. When the microspheres are arranged in an array, the
electromagnetic coupling between the microspheres, as shown in the illustration on the
left of Fig. R6 (a), will play an important role in determining the positions of the MD
and ED resonance peaks.

Here, we illustrate this by building an array of microspheres with an initial period of P_x
$= P_y = 300 \mu\text{m}$. Under the y -polarized incidence condition of the electromagnetic wave,
an increase in the array microsphere spacing P_x (P_y is fixed) induces a shift of the ED
resonance to lower frequencies, but the MD resonance is maintained at about 0.3250
207 THz. Conversely, when P_y is increased, the MD resonance undergoes a significant
redshift and the ED resonance peak remains stable at 0.4510 THz.

In order to reveal this phenomenon, we further simulated the E-field distribution of the
ED resonance (Fig. R6(b i) and Fig. R6(c i)) and the H-field distribution of the MD
resonance (Fig. R6(b ii) and Fig. R6(c ii)), which are used for comparing the changes
of the mutual coupling between the initial state and the microsphere spacing increase.
And it is found that the moments (i.e., local coupling strengths) of individual ED/MD
remain stable when the microsphere spacing is increased in either the x or y direction.
The main factor leading to the frequency shift of the ED/MD resonance can be
attributed to the mutual coupling between neighboring dipoles, which is dominated by
transverse coupling. Thus, by independently varying the microsphere spacing (P_x or P_y),
we can achieve independent control of the ED and MD resonance modes. A detailed
illustration of this mechanism can be found in the revised manuscript ("Bidirectional
strain detection strategy" section) and Supplementary Material Note S2.

 Fig. R6. (a) Schematic diagram of the ED and MD distribution in the array of dielectric microspheres.
 The inset on the left shows two coupling modes: transverse coupling (side-by-side alignment in blue
 dashed square) and longitudinal coupling (end-to-end alignment in red dashed square) between the
 dipoles. The inset on the right illustrates the unit cell parameters ($P_x = P_y = 300 \mu\text{m}$) and the
 boundary condition. Simulated resonance frequencies (including ED and MD resonances) versus
 cell period P_x (b) and P_y (c). The insets of (b) and (c) present the simulated E-field (i) and H-field
 (ii) distributions across the centre x - y plane of the microspheres as P_x (b)/ P_y (c) = $300 \mu\text{m}$ (left side
 of the dashed square) and P_x (b)/ P_y (c) = $460 \mu\text{m}$ (right side of the dashed square).

 3. What is the minimum strain that can be detected by the device, and what is the
 maximum strain can be applied to the device?

**Response:** Thank you for pointing out this essential problem in our manuscript. The
 simulated strain-resonance frequency curves (left figures of Figs. R7(a-b)) that are
 monotonically smooth and continuous, reveal that our device can theoretically detect
 an infinitesimal strain. But in fact, limited by the spectral resolution (0.001 THz) of
 the THz time-domain spectroscopy (TDS) device (QT-TRS1000, Quenda, China),
 the device indeed owns a minimum strain that can be accurately recognized.

According to the spectral resolution of the TDS system – 0.001 THz, as shown in insets
 of Figs. R7(a-b), the theoretically smallest strain values (1.25% @ x direction, and
 2.7% @ y direction) have been accurately determined, when the resonance shifts to
 lower frequency by 0.001 THz from the initial state where the applied strain is 0%.
 Then the corresponding test under the strain condition obtained above was conducted
 to observe the resonance frequency shifting. From Figs. R7(c-d), ~0.001-THz
 resonance frequency shifting can be implemented in the experiment when the x -
 directional and y -directional strains are respectively set to 1.25% and 2.7%, which are
 consistent with the simulated predictions (Figs. R7(a-b)). Corresponding description
 has been added in lines 251-257, page 11 of the revised manuscript as “In addition, the
 sensitivity of our meta-sensor has also been analyzed...”, and Figs. 3 (f-g, h, j).

In the end, we statistically analyze the above measured results. Trough 100-round test
 (Figs. R8(a-b)), the average resonance frequency and the standard deviation can be
 calculated as [0.4216 ± 0.0003 THz @ 0 strain along x direction; 0.4205 ± 0.0002
 254 THz @ 1.25% strain] (Fig. R8(a)) and [0.3221 ± 0.0004 THz @ 0% strain along y
 direction; 0.3210 ± 0.0005 THz @ 2.7% strain along y direction] (Fig. R8(b)).

 Fig. R7. Simulated resonance-strain relation of the proposed meta-sensor: (a) for ED resonance @
 x-directional strain; (b) for MD resonance @ y-directional strain. The insets of (a) and (b) present
 the simulated smallest strain. (c-d) Corresponding measured results for verifying the theoretically
 smallest strain.

 Fig. R8. Statistical analysis for the minimum strain that can be detected by our device: (a) for x-

directional strain; (b) for y -directional strain.

Due to our negligence, the maximum strain that can be applied to our sample has not
been analyzed in the previous manuscript. Following your thoughtful reminder, we
have finalized the strain limit (65% @ x direction, and 70% @ y direction) of our
design from the aspect of its mechanical feature and strain-resonance response.

For the mechanical stretchability, Fig. R9 depicts the measured stress-strain curves of
our sample, and it is obvious that our sample can be stretched with a mechanical fracture
strain larger than 150%. For the sensing performance, as indicated by the simulated
results in Fig. 3(d) of the revised manuscript, when the applied x -directional strain
exceeds 65%, the ED resonance will gradually become unrecognizable due to the
attenuation of the transmission amplitude, while the strain along the y direction can
reach 70% without interfering the MD resonance determination (Fig. 3(e) of the revised
paper). When the y -directional strain is larger than 70%, the originally stable ED
resonance related to the x -directional strain will shift to lower frequency (Fig. R10)
effecting the system calibration. This phenomenon can be attributed to the significant
enhancement of the longitudinal coupling between the adjacent EDs by comparing the
E-field distribution inside the white frames in insets i and ii of Fig. R10. Therefore, the
strain limit along y direction can be determined as 70%. Then, the corresponding strain
test was conducted, and the experimental results agree well with the simulated ones
(Figs. 3(d-g) of the revised paper) with the maximum external strain of 65% @ x
direction and 70% @ y direction, respectively. The corresponding description has been
added in lines 236-241, page 10 of the revise manuscript as "...Note that when the
applied strain exceeds ..." and the revised Fig. 3(c). In addition, Fig. R10 has been
added in the Supporting Information as Fig. S6.

Thanks again for your valuable suggestion. After detailed theoretical analysis and
experimental validation, the upper and lower detection limits of our sensors have
been finally determined as [65% @ x direction; 70% @ y direction] and [1.25%
@ x direction; 2.7% @ y direction], respectively.

Fig. R9. Measured strain-stress curves of the pure PDMS film and the fabricated sample.

Fig. R10. Simulated transmission spectra of meta-sensor arrays at 70% and 75% strain. Simulated
E-field distributions (@ corresponding ED resonances) across the centre x - y plane of the
microspheres as $\epsilon_y = 70\%$ (i) and $\epsilon_y = 75\%$ (ii).

**Reviewer #2 (Remarks to the Author):**

*The authors novelty applied Mie resonance to strain measurement and realized in plane*
*strain recognition. But the authors should complement the details of the strain sensing*
*and the applications. This manuscript could be accepted after fulfilling the experiments*
*and modifications mentioned below.*

**Response:** We appreciate the reviewer for the positive comments. We have carefully
addressed the reviewer's concerns. The manuscript and the Supplementary Information
have been revised accordingly.

*1. The data shown in figure 3e and 3f, figure 4c and 4d was not enough, each line in*
*those figures only possessed five points, which is not sufficient. The authors should*
*measure the resonance frequency under more different strains. And the authors should*
*measure the strain detection limit and the smallest strain variation that could change*
*the resonance frequency.*

**Response:** Thank you for your valuable suggestion. The ED and MD resonance
frequency shifting under more different strains have been comprehensively measured
to make the results much more convincing (see Figs. 3(d-g) of the revised paper).

Regarding the unidirectional tensile (*x*- or *y*-direction) strain experiments, as shown in
Figs. 3(d-g) of the revised manuscript, we have remeasured the transmission spectrum
information of the sample as **the applied strain increases by 5% each time (up to 65%**
**@ x direction and 70% @ y direction)**. Note that all the measured results agree well
with the simulated ones, demonstrating the correctness of our design method. For the
simultaneously bidirectional stretching experiments, **the strain test range has been**
**extended to 32% (strain step is set to 4%)** as shown in Figs. 4(a-d) of the revised
manuscript, and it's evident that the bi-directional strains can be independently detected
by our meta-sensor.

For the mechanical stretchability, the stress-strain curves of our sample were measured,
and it is obvious that our sample can be stretched with a mechanical fracture strain

larger than 150% (Fig. R11). For the sensing performance, as indicated by the simulated
results in Fig. 3(d) of the revised manuscript, when the applied x -directional strain
exceeds 65%, the ED resonance will gradually become unrecognizable due to the
attenuation of the transmission amplitude, while the strain along the y direction can
reach 70% without interfering the MD resonance determination (Fig. 3(e) of the revised
paper). When the y -directional strain is larger than 70%, the originally stable ED
resonance related to the x -directional strain will shift to lower frequency (Fig. R12)
effecting the system calibration. This phenomenon can be attributed to the significant
enhancement of the longitudinal coupling between the adjacent EDs by comparing the
E-field distribution inside the white frames in insets i and ii of Fig. R12. Therefore, the
strain limit along y direction can be determined as 70%. Then, the corresponding strain
test was conducted, and the experimental results agree well with the simulated ones
(Figs. 3(d-g) of the revised paper) with the maximum external strain of 65% @ x
direction and 70% @ y direction, respectively. **Therefore, the strain limit of our**
**design was finalized as 65% @ x direction and 70% @ y direction.**
The corresponding description has been added in lines 236-241, page 10 of the revised
manuscript as “...Note that when the applied strain exceeds ...”, and the revised Fig.
3(c). In addition, Fig. R12 has been added in the Supporting Information as Fig. S6.

Fig. R11. Measured strain-stress curves of the pure PDMS film and the fabricated sample.

Fig. R12. Simulated transmission spectra of meta-sensor arrays at 70% and 75% strain. Simulated
 E-field distributions (@ corresponding ED resonances) across the centre x - y plane of the
 microspheres as $\varepsilon_y=70\%$ (i) and $\varepsilon_y=75\%$ (ii).

Further, we have detailly uncovered the smallest strain variation that could change the
 resonance frequency. Considering that the simulated strain-resonance frequency curves
 (left figures of Figs. R13(a-b)) are monotonically smooth and continuous, our device
 theoretically has the capability to detect an ultra-small strain variation. However, in
 practice, confined by **the spectral resolution (0.001 THz) of the THz time-domain**
 **spectroscopy (TDS) device** (QT-TRS1000, Quenda, China), the smallest strain
 variation detection performance of our device is limited.

In principle, the smallest strain variation corresponds to the point on the curves with
 the steepest slope. Regarding the x -directional strain (Fig. R13(a)), the slope reaches its
 maximum value when the applied strain is equal to 65%. Conversely, strain-resonance
 frequency curve in Fig. R13(b) has the greatest slop with the externally y -directional
 strain of 0%. In the meantime, considering the spectral resolution of the TDS system
 (0.001 THz), the theoretically smallest strain variation value (**0.63% @ x direction**
 **and 2.7% @ y direction**) can be successfully obtained through shifting the resonance
 frequency by 0.001 THz towards the lower slop direction as shown in the insets of Figs.
 R13(a-b). Then the corresponding test under the strain condition obtained above was
 conducted to observe the resonance frequency shifting. From Figs. R13(c-d), ~0.001-
 383 THz resonance frequency shifting can be implemented in the experiment, which is
 384 consistent with the simulated predictions (Figs. R13(a-b)). We statistically analyze the
 385 above measured results. Trough 100-round test (Figs. R14(a-b)), the average resonance

frequency and the standard deviation can be calculated as [0.3511 THz/0.0002 THz @
 64.37% strain along x direction; 0.3500 THz/0.0004 THz @ 65% strain along x
 direction] (Fig. R14(a)) and [0.3221 THz/0.0004 THz @ 0 % strain along y direction;
 0.3210 THz/0.0005 THz @ 2.7% strain along y direction] (Fig. R14(b)). Therefore,
 the smallest strain variation that can be detected by our design was finalized as
 0.63% @ x direction and 2.7% @ y direction. Their difference can be attributed to
 the discrepancy of the corresponding resonance frequency shift bandwidth. The
 corresponding description has been added in lines 257-261, page 11 of the revise
 manuscript as “...Using the similar analysis approach...” and Figs. 3(f-g, i-j), and Note
 S5 in the Supporting Information. In addition, Fig. R14 has been added in the
 Supporting Information as Fig. S8.

 Fig. R13. Simulated resonance-strain relation of the proposed meta-sensor: (a) for ED resonance @
 x-directional strain; (b) for MD resonance @ y-directional strain. The insets of (a) and (b) present
 the simulated minimum strain variations. (c-d) Corresponding measured results for verifying the
 theoretically minimum strain variations.

 Fig. R14. Statistical analysis for the minimum strain variations that can be detected by our device:
 (a) for x-directional strain; (b) for y-directional strain.

2. As the authors mentioned in the abstract that arbitrary strain information could be
 detected. But the authors did not perform any experiments on the strain direction
 detection. The authors should design and perform more experiments to complement and
 support their conclusion.

**Response:** We gratefully appreciate your valuable comment. As your suggestion, the
 experiments on the strain direction detection have been conducted and the related
 results have been added in the revised manuscript. Fig. R15(a) shows the schematic and
 photograph of the strain direction detection system, where a right-angle fixture was
 adopted to fix our meta-senor and resist the correlative deformation induced by the
 Poisson's ratio of the PDMS substrate. In addition, the sample was fabricated as a
 quarter circle which can facilitate the calibration of the direction of the applied strain.
 In our experiment, five angular cases ($\theta = 15^\circ, 30^\circ, 45^\circ, 60^\circ, \text{ and } 75^\circ$) have been
 considered, and the corresponding ε_x and ε_y (Fig. R15(b)) have been successfully
 obtained with the increase of the external strain at different directions (from test round
 1 to 5). Finally, following the principle of orthogonal decomposition, the preset strain
 directions can be recognized and quantified through comparing the measured transverse

(ε_x) and longitudinal (ε_y) strains (Fig. R15(b)) according to the equation:
 $\theta = \arctan(\varepsilon_y / \varepsilon_x)$. The accurate test results shown in Fig. R15(c) are consistent with
 the theoretical expectations, fully demonstrating the strain direction detection ability of
 our design.
 The corresponding description has been added in lines 304-317, pages 13-14 of the
 revised manuscript as “...To further investigate the strain direction detection
 performance...”, and revised Figs. 4(e-g).

Fig. R15. (a) Schematic and photograph of the setup and the sample for conducting the strain
 direction detection. (b) ε_x and ε_y measured by our device that is applied with the external strains
 along the direction of $\theta = 15^\circ, 30^\circ, 45^\circ, 60^\circ,$ and 75° , respectively. (c) Calculated strain directions
 based on the measured ε_x and ε_y .

*3. Normally, mixing particles like ZrO_2 into PDMS would cause a significant decrease*
 *in mechanical properties. The authors should complement the experiments about*
 *mechanical properties.*

**Response:** Thank you very much for your valuable advice. According to your
 suggestion, the mechanical properties of the PDMS with and without ZrO_2 particles are
 tested by utilizing a universal testing system, and the corresponding stress-strain curves
 are provided in Fig. R16. In spite of a decrease in mechanical properties compared to
 pure PDMS ($\varepsilon = 243\%$), our device still exhibits a mechanically fractured tensile strain
 of $\varepsilon = 156\%$. Further, we measured the durability of our sensors. In addition, to further
 exhibit the cyclic tensile properties, we also have tested the transmission signal passing
 through the sample during 5000 tensile cycles as the applied strain repeatedly varies

from 0 to ~65% @ x direction and ~70% @ y direction. The resonance frequencies at
 strain of 0, ~65% @ x direction, and ~70% @ y direction almost unchanged (Fig. R17).
 The stable transmitted signals reflect the remarkable durability of the fabricated sample.
 The corresponding description has been added in lines 227-230 and 247-250, pages 10-
 11 of the revised manuscript as “Prior to investigating the strain sensing behavior...”
 and “...This meta-sensor also exhibits...”, and the revised Fig. 3(c). In addition, Fig.
 R17 has been added in the Supporting Information as Fig. S7.

 Fig. R16. Measured stress-strain curves of the pure PDMS film and the fabricated sample.

 Fig. R17. Measured ED and MD resonance frequencies of the fabricated sample after 1, 10, 50, 100,
 1000, 2000 and 5000 cycles of stretching (the applied strain repeatedly varies from 0 to ~65% @ x
 direction and from 0 to ~70% @ y direction): (a) for initial state (0% strain); (b) for ~65% strain
 along x direction; for ~70% strain along y direction.

 4. The authors declared that a sensor array containing ~1580 meta sensors was
 prepared, but in the manuscript, the authors did not exhibit any sensor array with 1580
 meta sensors. The authors only performed the application of the 6×6 sensor array. If
 this work could reach such a large number of sensors, the authors should design and

*perform more appropriate applications of the large sensor array.*

**Response:** We thank the referee for raising this point. Allow us to clarify the definition
of a meta-sensor and the calculation of approximately 1580 meta-sensors. In the
manuscript, a single meta-sensor is defined as a unit consisting of uniformly and
periodically dispersed ZrO₂ particles, with a size of ~3×3 mm² (approximately four
operating wavelengths) that is closed to the THz spot area for the subsequent testing
need. Then, a required meta-sensor array can be built by further expanding the ZrO₂
array to a large scale according to application requirement. In our paper, a 11×13 cm²
sample shown in Fig. 3(b) of the revised paper has been fabricated using our
manufacture method. Thus, the number of meta sensors contained in the sample can be
calculated by dividing the total area of the sample (11×13 cm²) by the area of a single
sensor (~3×3 mm²). The calculation results show that our sensor array contains 1580
periodically arranged meta-sensors. Please refer to the relevant figures and descriptions
in the revised manuscript for more details (Fig. 2(a); line 95, page 4 as “...containing
~1580 meta-sensors, each of which has a size of ~3 × 3 mm²...”).

Thank you again for your constructive comments. Although our array contains 1580
sensors, the current operable and monitoring area of the THz time-domain spectroscopy
(TDS) device is only 4 cm×4 cm. The current sensor arrays are still in the proof-of-
concept stage, and may still be far away from real practical applications. To verify its
potential applications, we utilize our sensor array for non-uniform in-plane strain
mapping. As shown in Fig. 5 of the revised manuscript, a 10×10 sensor array larger
than the 6×6 array in the last manuscript was demonstrated for strain mapping and our
test results are basically consistent with those measured using optical markers, verifying
the potential applications of our sensor array. More details can be seen in lines 327-351
on page 15 “In addition to these advantages...” and Fig. 5 in the revised manuscript.
We will also further improve our sensing system and explore its broader applications.

*5. In Figure 1 and Figure 3, the diagram shows that the structure of the sensor was the*

*ZrO₂ particles uniformly dispersed in the PDMS, but in Figure 5 the photograph shows*
*that the 6×6 sensor array presents a mesh-like structure. Is the sensor structure*
*changed in the sensor array?*

**Response:** Thanks for kind comment and we apologize for any confusion resulting
from the lack of clarity in our description. The meta-sensor designed in our paper is
indeed the ZrO₂ particles uniformly and periodically dispersed in the PDMS with a unit
size of 3×3 mm², and thus in our meta-sensor array (built by expanding the ZrO₂
array and substrate), each area with the same size can be regarded as a meta-
sensor. In Fig. 5 of the revised paper, to verify the potential application of our sensor
array for strain mapping, we used a marker to draw longitude and latitude lines to divide
the entire film into 100 areas. The strains in the 100 areas are measured using our meta-
sensors and calculated according to distance change of the mark lines for further 2D
strain mapping. To avoid possible confusion, we have added relevant descriptions in
line 95, page 4 as “...containing ~1580 meta-sensors, each of which has a size of ~3 ×
3 mm²...”

**Reviewer #3 (Remarks to the Author):**

*The manuscript “A terahertz meta-sensor array for 2D strain mapping” by X. Lu et al.*
*proposes a flexible meta-sensor array that can detect the in-plane direction and*
*magnitude of preloaded strains by referencing a dynamically transmitted terahertz*
*(THz) signal. Although terahertz metamaterial detection technology based on Mie*
*resonances is not new, this is an interesting attempt on a topic of current interest. Some*
*major concerns should be addressed to make the presentation more informative.*

**Response:** We appreciate the positive evaluation of Reviewer. To address all comments,
we conducted additional experiments and revised the manuscript and Supporting
Information.

*1. The idea of using dielectric particles to generate electric and magnetic resonances*
*were previously reported ([https://doi.org/10.1016/S1369-7021\(09\)70318-9](https://doi.org/10.1016/S1369-7021(09)70318-9);*
*<https://doi.org/10.1007/s00339-011-6716-2>). An efficient absorber in the THz band*
*with a monolayer of zirconium dioxide (ZrO₂) microspheres was also demonstrated*
*(<https://doi.org/10.1364/OE.26.013001>). But the authors haven't cited these references*
*in the manuscript.*

**Response:** Thanks for your kind comments. We are sorry that we did not cite these
highly relevant articles. We have cited these papers mentioned above as Refs. [34-36]
in the Introduction section of the revised manuscript as your suggestion.

34.Zhao, Q., Zhou, J., Zhang F. L., & Lippens, D. Mie resonance-based dielectric metamaterials. *Mater. Today* **12**, 60–69 (2009).

35.Kozlov, D. S., Odit, M. A., Vendik, I. B., Roh, Y. G., Cheon, S. & Lee, C. W. Tunable terahertz metamaterial based on resonant dielectric inclusions with disturbed Mie resonance. *Appl. Phys. A* **106**, 465–470 (2012).

36.Gao, J., Lan, C., Zhao, Q., Li, B. & Zhou, J. Experimental realization of Mie-resonance terahertz absorber by self-assembly method. *Opt. Express* **26**, 13001–13011 (2018)

*2. It is preferred to measure the strain sensor for wearable devices with a reflection*

*THz system instead of a transmission system, as THz signal cannot penetrate human*
*body, so the potential of this strain sensor for real applications is questionable.*

**Response:** We really appreciate for your constructive and insightful comments.
Although we have developed the transmission-type THz meta sensors for the detection
of strain magnitude and directions, it is indeed faced with the challenge as your
mentioned that THz signal can hardly penetrate human body. To address this issue, we
propose to design a reflection-type strain sensor through the similar strategy aroused in
our manuscript. Limited by the lack of reflection-type THz detection equipment, we
used simulations to confirm the feasibility of this hypothesis. The simulated results
indicate that the reflection-type meta strain sensor functions identically to the
transmissive one.

To validate our hypothesis, a reflection-type prototype was constructed, as shown in
Fig. R18(a) using the same design strategy as the transmission-type prototype described
in the main text. In this reflection-type design, a flexible reflection layer is adopted to
isolate the interference from the lower substrate, such as the human body. The initial
microspheres array period is $P_x = P_y = 300 \mu\text{m}$, and the distance between the full-
reflection layer and the microsphere array is optimized to $50 \mu\text{m}$.

In the following, we explored the effect of strain in the x - (ε_x) and y -directions (ε_y) on
the ED and MD resonances using the control variable method. As shown in Figs. R18(b-
c), an increase in ε_x (with ε_y fixed) leads to a downward shift in the ED resonance
frequency, while the MD resonance frequency remains at approximately 0.6460 THz.
Conversely, when ε_y increases with ε_x held constant, the MD resonance frequency
shows a redshift, while the ED resonance remains at ~ 0.7470 THz. These results
demonstrate that the reflection-type prototype exhibits similar behavior to the
transmission-type meta-sensor.

Fig. R18. Unidirectional tensile strain detection simulation performance evaluation of reflection-type meta-sensor. (a) Structure diagram of reflection-type meta-sensor. Simulated reflectance spectra of the meta-sensor array applied with different tensile strains (from 0 to 33.3%): (b) for the x -direction; (c) for the y -direction.

Fig. R19. Bidirectional strain detection performance evaluation of the reflection-type meta-sensor. Simulated reflectance spectra of the meta-sensor array simultaneously applied with x - and y -directional tensile strain: (a) y -directional tensile strain varying from 3.33% to 33.33% while the orthometric strain is maintained at 3.33%, 6.67%, 10%, 13.33%, 16.67%, 20%, 23.33%, 26.67%, 30%, and 33.33%; (b) y -directional tensile strain varying from 3.33% to 33.33% while the orthometric strain is maintained at 3.33%, 6.67%, 10%, 13.33%, 16.67%, 20%, 23.33%, 26.67%, 30%, and 33.33%.

Further, we conducted simulations for cases involving dual-strain loading, where strains
were applied simultaneously along both the x - and y -directions. As shown in Fig.
R19(a), when the x -directional strain varies from 3.33% to 33.33% companied with the

*y*-direction strains fixed at 3.33%, 6.67%, 10%, 13.33%, 16.67%, 20%, 23.33%,
26.67%, 30%, and 33.33%, respectively, the ED resonance gradually shifts to lower
frequencies as the *x*-directional strain increases. In contrast, the MD resonance remains
localized at specific values, only associated with the *y*-directional strain (3.33% @
0.6463 THz, 6.67% @ 0.6458 THz, 10% @ 0.6453 THz, 13.33% @ 0.6449 THz, 16.67%
@ 0.6443 THz, 20% @ 0.6436 THz, 23.33% @ 0.6429 THz, 26.67% @ 0.6419 THz,
30% @ 0.6410 THz, and 33.33% @ 0.6392 THz). By switching the strain loading mode,
the MD resonance frequency decreases with increasing strain in the *y*-direction (Fig.
R19(b)), while the ED resonance remains at specific values only related to the strain in
the *x*-direction (3.33% @ 0.7533THz, 6.67% @ 0.7492 THz, 10% @ 0.7445 THz,
13.33% @ 0.7391 THz, 16.67% @ 0.7336 THz, 20% @ 0.7274 THz, 23.33% @ 0.7208
596 THz, 26.67% @ 0.7144 THz, 30% @ 0.7069 THz, and 33.33% @ 0.6998 THz). **These**
597 **phenomena demonstrate that our proposed sensing mechanism can be directly**
**extended to the design of the reflection-type strain meta-sensor, making it much**
**more suitable for real applications, without compromising its basic performance.**
The aforementioned description all has been added into the Supporting Information as
Note S8 to highlight the scalability and generalizability of our design method.

Thanks again for your valuable comment. Although we have verified the feasibility of
the reflection-type strain meta-sensor by simulation, we regret to inform you that due
to limitations in our experimental conditions, we are unable to experimentally verify
the feasibility of reflection-type strain meta-sensor, which will be a focus of our future
research.

*3. In page 9, the authors described a strategy for fabrication of the metasensor array,*
*noticed the diameter of ZrO₂ microsphere is 161 um and the thickness of PDMS*
*substrate is 320 um, please clarify how to ensure that the microballs were distributed*
*on the same horizontal plane embedded in the substrate. How to avoid frequency shift*
*caused by different positions of microballs?*

**Response:** Thanks for your valuable comment. The meta sensor array was fabricated
through a multi-step method. To ensure the microballs distributed on the same
horizontal plane in the PDMS matrix, an even and flat supporting PDMS layer was first
fabricated to calibrate the reference surface.

Fig. R20. Schematic of the fabrication process for the meta-sensor array.

To obtain the supporting PDMS layer, liquid PDMS is scraped onto a highly precise
platform by a high precision scraper (HQ-TB-G, Huaqi Instrument Co., LTD, China)
with the accuracy of $\pm 0.5 \mu\text{m}$ (STEP 1 of Fig. R20). Then, the PDMS liquid film on the
platform was pre-cured gradually in an oven. After that, an even and flat pre-cured
PDMS film with the size of $10 \times 10 \text{ cm}^2$ can be implemented with the maximum
fluctuation less than $0.9 \mu\text{m}$ within 1 cm range (Fig. R21).

Fig. R21. Evaluation of the surface flatness of the support layer. (a) Optical photographs of the
 samples. Test positions are marked with red lines in the figure. (b-j) Scan results of surface
 conditions at the sample marker location.

Then, the ZrO_2 microspheres were precisely arrayed onto the supporting PDMS layer
 using a highly accurate sieve mesh, ensuring that they are nearly on the same horizontal
 plane (STEP 2 in Fig. R20). Finally, the meta-sensor array was prepared by
 encapsulating the ZrO_2 microsphere array with the PDMS (STEP 3 in Fig. R20). As
 shown in the SEM images, the microspheres are nearly on the same horizontal plane
 (Fig. R22). Because the fluctuations of the supporting PDMS layer (no more than 0.9
 640 μm) is much smaller than the ZrO_2 microsphere size, in principle little frequency shift
 will be caused by the different positions of microballs.

Fig. R22. SEM image of the section of the meta-sensor sample.

To confirm this, we measured the resonance-strain response at the different locations of the fabricated sample (the six red points shown in Fig. R23(a)). As observed, the MD and ED resonance peaks at various positions on the sample are almost the same when subjected to the same strain in x - (Fig. R23(b)) and y -direction (Fig. R23(c)) during tensile tests. This proves that the microsphere position deviation caused by the support layer has little effect on the overall performance of the device.

The above reached conclusion has been added into Methods section of the revised manuscript as “...With our method, the ZrO₂ microspheres are nearly on the same horizontal plane...” (lines 393-396, page 17), and Fig. R23 also has been added into the Supporting Information as Fig. S15.

Fig. R23. (a) Optical photographs of the sample tested with the test positions marked with red dots.

Transmission spectra of the sample at different positions during stretching in the x-direction (b) and

y-direction (c).

4. In Fig.3c, please describe why does the tensile force along x-direction not affect the

frequency shift of the peak between ED and MD resonances, but it is observed in the

simulation results? And please clarify why the parameter of the tensile strain is only set

to 28%. What would happen if it is set higher?

**Response:** We really appreciate the referee's comments. In your third comment, we

have proved that the layer (blue color in Fig. R24(a)) for supporting the ZrO_2 particles

are very even and flat, which can help the particle array distributed on the almost same

horizontal plane and shows little effect on the resonance frequency shift. Therefore, the

phenomenon as your concerned we think maybe induced by the uneven thickness of

encapsulation layer (gray color in Fig. R24(a)).

To prove our hypothesis, we investigated the effect of the thickness variation of the
encapsulation layer on the peak frequency between the ED and MD resonances. As
shown in Figs. R24(b-f), two patterns of change can be observed:

i. With the increase of the applied strain (from Figs. R24(b) to (f)), your concerned
peak frequency will gradually shift to lower frequency.

ii. Under the identical strain, a thinner encapsulation layer will lead to the blueshift
of the corresponding peak frequency (insets of Figs. R24(b) to (f)).

Therefore, comprehensively considering the above two laws that mutually constrain
each other, if a greater strain is applied to a device owning a thinner encapsulation layer,
the peak frequency between ED and MD resonances in principle will keep stable. Based
on this finding, an individual case has been raised in Fig. R24(g), where the similar
phenomenon as what you previously observed can be revealed. **Therefore, the uneven**
**thickness of encapsulation layer is the source affecting the frequency shift of the**
**peak between ED and MD resonances.** To address this problem, when reconduct the
test, the test point on the sample was first marked to make sure that the tensile response
was always measured at the same location. In addition, the encapsulation layer was
optimized during the preparation process, and thus the peak between ED and MD
resonances occurs expected movements as shown in Fig. 3(d) of the revised paper.

In addition, it is worth to note that the fixed MD resonance frequency and the stable ED
resonance variation range observed in above analysis (Figs. R24(b-f)) prove the high
robustness of our strain detection approach.

Fig. R24. Sample model structure diagram (blue for support layer, gray for encapsulation layer).

(b-d) Simulation of the effect of encapsulation thickness on resonance peak (between MD and

ED) under different strain variables. (b) $\epsilon_x=0$; (c) $\epsilon_x=7\%$; (d) $\epsilon_x=14\%$; (e) $\epsilon_x=21\%$; (f) $\epsilon_x=28\%$; (g) A

case obtained by extracting the spectrum in (b-f).

Due to our negligence, the maximum strain that can be applied to our sample has not
 been analyzed in the previous manuscript. As your kind reminder, we finalize the strain
 limit (**65% @ x direction, and 70% @ y direction**) of our design from the aspect of
 its mechanical feature and strain-resonance response.

For the mechanical stretchability, the stress-strain curves of our sample were measured,
 and it is obvious that our sample can be stretched with a mechanical fracture strain
 larger than 150% (Fig. R25). For the sensing performance, as indicated by the simulated
 results in Fig. 3(d) of the revised manuscript, when the applied x -directional strain
 exceeds 65%, the ED resonance will gradually become unrecognizable due to the
 attenuation of the transmission amplitude, while the strain along the y direction can
 reach 70% without interfering the MD resonance determination (Fig. 3(e) of the revised
 paper). When the y -directional strain is larger than 70%, the originally stable ED
 resonance related to the x -directional strain will shift to lower frequency (Fig. R26)
 effecting the system calibration. This phenomenon can be attributed to the significant

enhancement of the longitudinal coupling between the adjacent EDs by comparing the
 E-field distribution inside the white frames in insets i and ii of Fig. R26. Therefore, the
 strain limit along y direction can be determined as 70%. Then, the corresponding strain
 test was conducted, and the experimental results agree well with the simulated ones
 (Figs. 3(d-g) of the revised paper) with the maximum external strain of 65% @ x
 direction and 70% @ y direction, respectively. (The corresponding description has been
 added in lines 236-241, page 10 of the revised manuscript as “...Note that when the
 applied strain exceeds ...”, and the revised Fig. 3(c). In addition, Fig. R26 has been
 added in the Supporting Information as Fig. S6.)

**Therefore, our device can work normally when the tensile strain is greater than**
 **28% but not exceed the threshold values – 65% @ x direction and 70% @ y**
 **direction.**

Fig. R25. Measured stress-strain curves of the pure PDMS film and the fabricated sample.

Fig. R26. Simulated transmission spectra of meta-sensor arrays at 70% and 75% strain. Simulated

E-field distributions (@ corresponding ED resonances) across the centre x - y plane of the
microspheres as $\varepsilon_y=70\%$ (i) and $\varepsilon_y=75\%$ (ii).

*5. Please provide additional data for y -directional tensile strain with 28% in Fig 4a*
*and x -directional tensile strain with 28% in Fig 4b, which can make the results more*
*convincing. And according to the author's description, are these data derived from an*
*average of 5000 times? Please add scale bar in Fig. 5. Similarly, please provide the*
*relevant data when the stress level reaches 28% according to the authors strategy.*

**Response:** Thank you very much for your kind comments.

According to your suggestion, the maximum strain applied in the bidirectional strain
measurement has been increased to 32% covering the original 28%, and the
corresponding results have been presented in Figs. 4(a-b) of the revised paper. Based
on the resonance information extracted from Figs. 4(c-d) of the revised paper, it is easy
to find that the proposed meta-sensor can independently recognize and quantify the
external strains for orthogonal loading. This further validates the correctness of our
design strategy.

We apologize for any confusion caused by our previous description. In fact, each THz
signal was collected **100 times** per test round and six sets of parallel tests were
conducted for calculating the average value and determining the corresponding
standard deviation. We have clarified it in lines 406-408, page 17 of the revised
manuscript as "...When conduct the measurement, each THz signal was collected 100
759 times per test round for calculating the average value as well as six sets of parallel tests
for determining the corresponding standard deviation."

As your valuable suggestion, a new 2D strain distribution has been reloaded to our
device, of which the stress level is more than 28%. The corresponding measured results
have been illustrated in new Fig. 5 of the revised manuscript, which reveals fine
discrepancy when compared to the visual computation analysis. In addition, the
necessary scale bars also have been embedded into Fig. 5 with the help of your reminder.

6. The key parameters of a strain sensor include high stretchability and high sensitivity
(gauge factor). In the manuscript, the measured strain step is 7% (Fig. 3), which
actually is too large for a strain sensor. The authors should calculate the gauge factor
of their design. The stretchability of this manuscript is below 30%, which is lower than
the reported strain sensor using the same material-PDMS
(<https://doi.org/10.1021/nn501204t>). An ultra-stretchable and highly sensitive strain
sensor has been reported with an ultra-high stretchability of 1000% and high sensitivity
with a maximum gauge factor (GF) of up to 165 (DOI
<https://doi.org/10.1039/C8NR08589G>). The authors should make a comparison with
these results.

**Response:** Thank you very much for your suggestion. According to your kind reminder,
we have retested the resonance-strain response of the fabricated meta-sensor under the
conditions – **x direction: varying from 0 to 65% by 5% per step and y direction:**
**varying from 0 to 70% by 5% per step** (shown in Figs. 3(d-e) of the revised
manuscript), remarkably enriching the experimental content. Considering the smallest
strain values (1.25% @ x direction, and 2.7% @ y direction) that can be detected by our
device (shown in Figs. 3(h) and (j) of the revised paper), the strain step is set to 5%.
The corresponding measured results are presented in Figs. 3(d-f), which shows the
highly consistent with the simulated ones (Figs. 3(d-f)).
In your fourth comment, we have uncovered the maximum strain (**65% @ x direction,**
**and 70% @ y direction**) that can be detected by our device. To further prove the
excellent cyclic tensile properties, we also have tested the transmission signal passing
through the sample through 5000 tensile cycles as the applied strain repeatedly varies
from 0 to ~65% @ x direction/~70% @ y direction, and the corresponding results can
be found in Fig. R27 (has been added in the Supporting Information as Fig. S7). The
stable transmitted signals reflect the remarkable durability of the fabricated sample and
prove that **our stretchability is evenly matched with the design (up to 70%) in Ref.**
**[1] as your mentioned.**

Fig. R27. Measured ED and MD resonance frequencies of the fabricated sample after 1, 10, 50, 100, 1000, 2000 and 5000 cycles of stretching (the applied strain repeatedly varies from 0 to ~65% @ x direction and from 0 to ~70% @ y direction): (a) for initial state (0% strain); (b) for ~65% strain along x direction; for ~70% strain along y direction.

Then, as your valuable suggestion, we have calculated the gauge factor (GF) of our design to uncover its sensitive. GF is usually used for piezoresistive strain sensor, which is the ratio of relative change in electrical resistance to the mechanical strain.

Accordingly, in our case, we defined the GF as

$$GF = \frac{\Delta f}{f_0 \Delta \varepsilon} \quad (R1)$$

where Δf is the resonance frequency shift value from the initial resonance frequency f_0 , and $\Delta \varepsilon$ denotes the applied strain. Then, by substituting the measured results (Figs. 3f-g of the revised manuscript) into Eq. (R1), strain- $\Delta f/f_0$ curves and the corresponding fitted ones can be obtained as shown in Fig. R28. Following, the maximum slopes of the fitted curves, which correspond to the maximum GF , are calculated to be ~0.413 @ x direction and ~0.09 @ y direction.

The above description about the GF of our meta sensors has been added in lines 264-269, page 11 of the revised manuscript as "...Further, the maximum gauge factor (GF)..." and Fig. 3(k). In addition, the detailed process for determining the GF has been supplied in the Supporting Information as Note S6.

Although compared with the strain sensor reported in Ref. [1-2], the GF and stretchability of our design is much lower (Table R5), our proposed meta-sensor shows unique superiority in independently bi-directional strain detection, 2D strain mapping and strain direction recognition, which by now is still full of

823 challenge. In future work, we will further improve our sensor to enhance its Gauge
 factor and stretchability.

Fig. R28. Relative resonance frequency variation ($\Delta f/f_0$)-strain relationship.

Table R5. Comparison of our meta-sensor array with the proposed strain sensors.

Device type	Detection range	Orientation recognition	GF	References
Resistive-type	70%	×	2~14	[1]
Capacitive-type	1000%	×	165	[2]
Mie resonance	x-direction:65% y-direction:70%	✓	x-direction:0.413 y-direction:0.09	This work

1. Amjadi M, Pichitpajongkit A, Lee S, et al. Highly stretchable and sensitive strain sensor based on
 silver nanowire–elastomer nanocomposite[J]. *ACS Nano*, 2014, **8**(5): 5154-5163.

2. Xu H, Lv Y, Qiu D, et al. An ultra-stretchable, highly sensitive and biocompatible capacitive
 strain sensor from an ionic nanocomposite for on-skin monitoring[J]. *Nanoscale*, 2019, **11**(4): 1570-
 1578.

**Reviewer #4 (Remarks to the Author):**

*In this manuscript, the authors proposed a simple and unique meta-sensor array that*
*can detect the strains by referring to the dynamically transmitted terahertz (THz) signal.*
*The strain sensing mechanism based on Mie resonance is discussed in detail and the*
*correspondence between the electric/magnetic dipole resonance frequency and the*
*horizontal/vertical tension level is established. The information on strain magnitude*
*and direction at any position in the plane can be obtained through the proposed strategy,*
*as demonstrated by the experimental results. In addition, they also demonstrate the*
*unique large-area preparation process and self-cleaning function of meta-sensor array.*
*Overall, the paper is very nicely written and well structured. Considering the*
*significance and broad-ranging applications of the developed strategy, I would*
*recommend its publication in Nature Communications, after the following minor issues*
*are addressed.*

**Response:** We appreciate the reviewer for the positive comments. We have carefully
addressed the reviewer's concerns. The manuscript and the Supplementary Information
have been revised accordingly.

*1. In the experiment, the zirconia (ZrO₂) microsphere array in a polydimethylsiloxane*
*(PDMS) substrate was utilized to prove the proposed strain sensing mechanism, while*
*in the "Bidirectional strain detection strategy" part, the sensing mechanism is*
*established in a vacuum background. Thus, the authors should discuss the effect of*
*PDMS on electric/magnetic dipole resonance and the spectral information of pure*
*PDMS in the THz band should be supplemented.*

**Response:** Thank for your valuable comment. To uncover the effect of the substrate
PDMS on the ED and MD resonances, we introduced PDMS into the initial ZrO₂
microsphere array (Fig. R29(a)) for simulation that was used for establishing the basic
sensing mechanism. As shown in Fig. R29, the PDMS is gradually added towards two
opposite directions from the reference plane (dot line shown in Fig. R29) at the center
of the ZrO₂ sphere. When Δh increases from 0 to 260 μm , three stages can be observed

from in the shift of the ED and MD resonance.

Fig. R29. (a) Schematic diagram of the ZrO_2 microsphere array in the vacuum. (b) Schematic for
 analyzing the effect on the ED and MD resonances. (c-e) Simulated transmittance spectra of the
 meta-sensor in terms of different Δh . Simulated E-field (i) and H-field (ii) distributions across the
 center of the microspheres: (f) for $\Delta h = 0 \mu\text{m}$; (f) for $\Delta h = 80 \mu\text{m}$; (f) for $\Delta h = 100 \mu\text{m}$; (f) for $\Delta h =$
 $260 \mu\text{m}$.

**Stage 1:** the ED and MD resonances exhibit a significant shift (Fig. R29(c)), when the
 PDMS increases from 0 to the point where it wraps around the ZrO_2 sphere just right
 (namely Δh is equal to $80 \mu\text{m}$). During this process, the MD resonance gradually moves
 to higher frequency while the ED resonance shifts to the opposite direction. This can
 be attributed to the decayed ED and MD amplitude of the ZrO_2 microsphere array when
 embedded in the PDMS substrate (Fig. R29(g)) compared to that in the vacuum
 background (Fig. R29(f)).

**Stage 2:** due to the little variation in E-field and H-field distribution (Figs. R29(g) and
 (h)), the ED resonance only appears a slight shift (Fig. R29(d)), while the MD resonance
 remains relatively stable when Δh increases from 80 to $100 \mu\text{m}$.

**Stage 3:** when Δh is larger than $100 \mu\text{m}$, the ED and MD resonance frequency keep

stable (Fig. R29(e)) as a result of the almost same E-field and H-field distributions (Figs.
R29(h) and (i)). In addition, with the increase of Δh , more THz energy is consumed
through the dielectric loss of the PDMS substrate. Consequently, fewer E-field can be
observed localized in the PDMS substrate (indicated by white wireframes in Figs.
R29(h-i)). That's the reason why the transmittance at the ED resonance gradually
decays as Δh increases. The corresponding description has been added into the
Supporting Information as Note S3.

According to your suggestions, we have also measured the spectral information of pure
PDMS in the THz band. We have included the spectral information of a pure PDMS
film with an identical thickness to that of our fabricated sample in Fig. R30. As shown
in Fig. R30, the pure PDMS film shows high and stable transmittance even under
stretchable state, consisting well with its intrinsic material property. Fig. R30 has been
added into the Supporting Information as Fig. S3.

Fig. R30. Measured transmission spectra of the pure PDMS under different external strains.

2. The meta-sensor array proposed in the manuscript has a sensor density of $\sim 11.1 \text{ cm}^{-2}$,
please explain the origin of this value. Since the proposed meta-sensor array is not
the same as the conventional electrical sensor array, how do the authors define the
"sensing density" of this meta-sensor array?

**Response:** We thank the referee for the careful reading and pointing out this issue. In
our paper, comprehensively considering the THz spot area for testing and the maximum

detection resolution, the size of a single meta-sensor is set to $3 \times 3 \text{ mm}^2$ (about 4
operation wavelengths). Based on this, we have calculated the sensor density of our
meta-sensor array to be approximately $\sim 11.1 \text{ cm}^{-2}$, following a methodology similar to
that used for traditional sensor density calculations.

*3. In Supplementary Fig. S4, SEM image of the section of the meta-sensor array is*
*presented. However, the microsphere arrays are not positioned in the middle layer of*
*the encapsulation layer. The reviewer would like to know if the thickness of the PDMS*
*encapsulation layer film has any effect on the device performance?*

**Response:** We really appreciate the referee's comments. To explore this issue raised
by the reviewer, we have analyzed the effect of the thickness of the encapsulation layer
(h_1 in Fig. R31(a)) on the device performance from the aspect of simulation. From the
first figures in Figs. R31(b-c), we can find that the initial ED and MD resonance
frequencies (without external strain applied) remain stable even the thickness of the
encapsulation layer varies from $180 \mu\text{m}$ to $300 \mu\text{m}$. When subjected to an identical x -
directional strain (from 14% to 70%), the corresponding MD resonance shifts to lower
frequency as h_1 gradually increases, while the ED resonance is always stable (Fig.
R31(b)), conforming to the requirement of a strain sensor. In the meantime, a similar
phenomenon also can be found in Fig. R31(c) in terms of the external y direction strain.
In other words, **the thickness of the PDMS encapsulation layer only affects** the
frequency shifts of the ED and MD resonances induced by strain. Specifically, a greater
thickness results in larger frequency shifts. However, this variation in thickness does
not disrupt the overall performance of the device.

Fig. R31. The effects of the encapsulation layer on the performance of the meta sensors. (a)
 Schematic representation of the model, where the encapsulation layer is shown in dark blue.
 Simulated transmittance spectra of the meta-sensor as h_1 varies from 180 μm to 300 μm : (b) for x -
 directional strain from 0 to 70%; (c) for y -directional strain from 0 to 70%.

4. In Supplementary Fig. S6, please provide more information on the cyclic stability
 during strain, not just after the release of the stretch cycle.

Response: Thank you very much for your kind suggestion. Accordingly, we have
 942 repeated the cyclic tensile test, and measured the transmission information of our device
 under stretched states with the strain of 65% @ x direction and 70% @ y direction)
 during 5000 tensile cycles (Figs. 32(b-c)). The resonance frequencies at strain of 0, ~65%

@ x direction, and $\sim 70\%$ @ y direction almost unchanged. The stable transmitted
 signals reflect the remarkable durability of our meta sensors. The corresponding
 description has been added in lines 247-250, pages 10-11 of the revise manuscript as
 “...This meta-sensor also exhibits excellent...”, and the new figures (Fig. R32) have
 replaced the original ones in the Supporting Information Fig. S7.

**Fig. R32.** Measured ED and MD resonance frequencies of the fabricated sample after 1, 10, 50, 100,
 1000, 2000 and 5000 cycles of stretching (the applied strain repeatedly varies from 0 to $\sim 65\%$ @ x
 direction and from 0 to $\sim 70\%$ @ y direction): (a) for initial state (0% strain); (b) for $\sim 65\%$ strain
 along x direction; for $\sim 70\%$ strain along y direction.

*5. Do superhydrophobic micro-nano structures have an effect on resonance signals?*
 *Please provide relevant explanations.*

**Response:** Thank you very much for your comment. The hydrophobic layer is made of
 PDMS material identical to the substrate of our device, and the SEM cross-section of
 the sample (Fig. R33) shows that it is composed of circular truncated cones with a
 diameter of $\sim 5 \mu\text{m}$ and a height of $\sim 5 \mu\text{m}$, which is far smaller than the operation
 wavelength ($0.7\sim 1 \text{ mm}$) of our device and cannot interact with the incident THz waves.
 We further investigated the ED and MD resonance frequency shifts of the normal meta
 sensors and superhydrophobic ones under various strains by simulation (Fig. R34). The
 results shows that the transmission curves of normal meta-sensors and
 superhydrophobic ones basically overlap, indicating that the hydrophobic layer has
 almost no effect on the performance of our meta sensors.

Fig. R33. Characterization of the meta-sensor array samples with “self-cleaning effect”.

Fig. R34. Simulation results of the influence of samples with superhydrophobic layers on device performance compared with normal samples.

6. In Video S3, Video S4, please add some captions to make it easier for the reader to understand.

Response: Thanks for the kind suggestion. Accordingly, we have added subtitle explanations in Video S3 and S4.

REVIEWER COMMENTS

Reviewer #1 (Remarks to the Author):

The authors have meticulously crafted a comprehensive response to both my comments and those provided by other reviewers. A substantial number of technical concerns have been effectively addressed. Nonetheless, certain issues require major revision before the manuscript can be considered for publication in Nature Communications.

These specific concerns are outlined below:

1. Despite the substantial volume of work and data presented in this manuscript, I harbor reservations regarding its novelty. In the rebuttal file, the authors emphasize their distinctive operational performance and fabrication method. While acknowledging an improvement in performance compared to the Advanced Optical Materials paper, the pivotal question revolves around whether this technological enhancement sufficiently justifies publication in Nature Communications. I am inclined to believe that the manuscript falls short of achieving a groundbreaking milestone, given its adherence to the same working principle as outlined in the Advanced Optical Materials paper. Regarding the fabrication method, the creation of a large-area flexible metamaterial through nanotransfer printing has been extensively explored and documented years ago (Nature Nanotechnology volume 6, pages 402–407 (2011)), making it a well-established concept rather than a novel contribution. In any case, I recommend that the authors delve further into elucidating the innovative aspects of their work.

2. The authors claim that the proposed sensor possesses “shows great potential for future applications in the sensing layer of flexible IoTs and wearable devices.”. In fact, all data in the manuscript come from ideal test setups (Supplementary Fig. S16). Besides, it is worth noting that the bulky 2D THz scanning platform must be taken into consideration when it is used for practical application. It's hard to imagine how the author could implement the sensor into a wearable device. It is recommended that authors describe the suitable application scenarios in their manuscripts.

3. In practical scenarios, particularly in wearable applications they claimed, out-of-plane interference in the z-axis direction is typically inevitable, particularly given the flexibility of their substrate, making them susceptible to such deformations. Such interference may manifest as bending and distortion. Notably, the authors exclusively investigated in-plane deformations. It is advisable for the authors to augment their study by incorporating simulation or test data that considers the sensor's response to bending and twisting deformations.

4. The content provided in their response document was not comprehensively included in the supplementary materials. It is strongly recommended that the authors integrate References [1-3] into the Comparison table within the supplementary material.

Reviewer #2 (Remarks to the Author):

The authors have done a lot of work to address the review's comments, I think the MS could be accepted for publication in this version.

Reviewer #3 (Remarks to the Author):

The author's response to question 2 is unsatisfactory. It is evident that the author stated the proposed scheme's application scenario is a wearable device, indicating that only an experimental plan utilizing the reflection system can accomplish this objective. However, the author's reply indicates that the lack of experimental equipment restricts the reflection scheme to simulation only. However, based on my understanding, reflection and transmission geometry are the fundamental optical path structures in terahertz systems and can be flexibly adapted. Moreover, numerical calculation in real-world for a reflection system is more challenging. If there is only simulation data available without any experimental data, it implies that the proposed scheme may not be feasible for the intended goal or merely suggests that future schemes based on this proposal could be beneficial for future applications.

Reviewer #4 (Remarks to the Author):

The authors took into account my comments. In my opinion, the article has become better and can be published in the journal Nature Communications.

**Point-by-point response to reviewer:**

**Reviewer #1 (Remarks to the Author):**

*The authors have meticulously crafted a comprehensive response to both my comments*
*and those provided by other reviewers. A substantial number of technical concerns have*
*been effectively addressed. Nonetheless, certain issues require major revision before*
*the manuscript can be considered for publication in Nature Communications.*

**Response:** Thank you very much for your positive feedback and affirmation of our
previous reply. We have carefully addressed the reviewer's comments below.

*These specific concerns are outlined below:*

1. *Despite the substantial volume of work and data presented in this manuscript, I*
*harbor reservations regarding its novelty. In the rebuttal file, the authors emphasize*
*their distinctive operational performance and fabrication method. While*
*acknowledging an improvement in performance compared to the Advanced Optical*
*Materials paper, the pivotal question revolves around whether this technological*
*enhancement sufficiently justifies publication in Nature Communications. I am inclined*
*to believe that the manuscript falls short of achieving a groundbreaking milestone,*
*given its adherence to the same working principle as outlined in the Advanced Optical*
*Materials paper. Regarding the fabrication method, the creation of a large-area*
*flexible metamaterial through nanotransfer printing has been extensively explored and*
*documented years ago (Nature Nanotechnology volume 6, pages 402–407 (2011)),*
*making it a well-established concept rather than a novel contribution. In any case, I*
*recommend that the authors delve further into elucidating the innovative aspects of*
*their work.*

**Response:** Thank you very much for your comments. Actually, the working principles
of our meta sensors are completely different from that reported in Ref. [R1]. We will
detailly introduce the differences from the aspect of resonance type, motivation method,
and tuning mechanism:

**(1) Resonance type and motivation method**

In Ref. [R1], the parabolic-shaped meta-atom (Fig. R1(a)) is **essentially an electric**
**resonator with polarization dependence**. As shown in Fig. R1(a), when interacted
with the y -polarized incident wave (TM mode), **E-field will be strongly localized**
**between the y -directionally adjacent meta-atoms**, resulting in the generation of an
electric resonance (yellow line). According to the similar motivation method, another
electric resonance (blue line) can be motivated under the x -polarized incident wave (TE
mode) inducing **the intensively localized E-field between the x -directionally**
**adjacent meta-atoms**.

However, in our design, **E- and H-fields (at different frequencies) are strongly**
**localized inside the ZrO_2 microspheres (Fig. 2(b) in the revised manuscript)**. And
**a couple of Mie resonances including electric dipole (ED) and magnetic dipole (MD)**
**resonances can be simultaneously generated** when illuminated with a x - or y -
polarized incident THz wave (Fig. 2(a) in the revised manuscript). Since there is no
need to adjust the polarization state of the THz source, our principle is conducive to
improving the detection efficiency.

(2) Tuning mechanism

Due to the different resonance prototypes, their tuning mechanisms are significantly
different. For the sensor proposed in Ref. [R1], a x - or y -directional strain will
accordingly change the w value and l value of the meta-atoms, leading to a change in
**capacitance between the adjacent metamaterial resonators** (Eq. (6) in Ref. [R1]).
Based on the classical **RLC circuit tuning principle**, the TM-mode (yellow line in Fig.
R1(b)) and TE mode excited electric resonances (blue line in Fig. R1(c)) can be tuned
referring to the varying capacitance. (the detailed introduction of the working principle
can be found in page 2 of Ref. [R1])

However, in our design, the initial ED and MD resonances can be independently tuned
**by changing the transverse coupling degree of EDs/MDs which is directly related**
**to the x -/ y -directional spacing between the adjacent ZrO_2 particles (Figs. 2(d-e) in the**
**revised manuscript)**. (the detailed introduction of the working principle can be found in
pages 5-8 of the revised manuscript)

Finally, the detailed comparison of the working principles between our work and that in Ref. [R1] has been concluded in Table R1. It is evident that the resonance type, motivation method, and tuning mechanism of our meta-sensor for detecting the external strain are distinctly different from that reported in Ref. [R1].

Table R1. Detailed comparison of the working principle

Reference	[R1]	This work
Resonance type	Electric resonance	ED and MD resonances
Resonance location	Between the adjacent meta-atoms	Inside the meta-atoms
Motivation method	E-field localization	E- and H-field localization
Tuning mechanism	RLC circuit tuning principle	Transverse coupling
Polarized-dependence	Yes	No

In addition, it is worthy to note that our tuning mechanism is more robust in bi-directional strain detection compared to that in Ref. [R1]. Regarding that in Ref. [R1], as shown in Figs. R1(b-c), when applied with a x/y -directional strain, the varied w value and l value will simultaneously affect the transverse and longitudinal capacitance between the adjacent meta-atoms, thereby leading to the crosstalk of the TE-wave- (yellow line) and TM-wave induced (blue line) resonance frequencies. However, for our design, the transverse coupling is dominant to affect the working frequency, e.g., once the ED/MD resonance is tuned, the MD/ED resonance always remains stable (as shown in Figs. 2(d-e) of the revised manuscript). Based on this advantage, our meta-sensor facilitates much more complex and practical functionalities, such as strain direction identification and 2D strain mapping than that in Ref. [R1].

Fig. R1. (a) Schematic illustrations of the proposed PSM device (reported in Ref. [R1]) operated by

80 applying a stretching force along the x - and y -axis directions. Experimental results and optical
microscope images of a PSM device stretched with different w (b) and l (c) values in the TE and
TM modes. Inserted scale bars are $50\ \mu\text{m}$.

The nanotransfer printing method reported in Ref. [R2] needs a **complicated**
**semiconductor process** (as shown in Fig. R2) and is strongly dependent on **advanced**
**semiconductor processing equipment** including high-precision PECVD machine,
deep ultraviolet projection mode photolithography, ICP-RIE etcher, and Nano-
imprinter, which are always **highly cost**. In addition, these steps shown in Fig. R2 call
for highly technological and operating level, favoring high-precision micro- and nano-
structures, but the **low yield rate** is still a common-recognized challenge. However, our
strategy only requires a **simple three-step process** -- “**scraping-array-encapsulation**”,
which can be flexibly achieved only by adopting a **scraper** (STEP 1 of Fig. R3) and a
**sieve screen** (STEP 2 of Fig. R3). Only simple curing operations are utilized throughout
the process (Fig. R3), making our method **highly robust**. **Therefore, our method has**
**great advantages in low-cost and large-area preparation.** (the detailed introduction
of our fabrication method can be found in “Method” section of the revised manuscript
and Note S4 of the Supporting Information)

Additionally, it is worthy to note that in Ref. [R2], the printing process involves using
diluted HF to remove the sacrificial SiO_2 layer. However, in our manufacturing strategy,
no additional chemical reagents (other than PDMS) are required throughout the entire
process, **which is more environmentally friendly and safer.**

Fig. R2. Schematics showing complete steps from stamp fabrication, multi-layer growth to transfer printing. (reported in Ref. [R2]).

Fig. R3. Schematic of the fabrication process for the meta-sensor array.

Reference:

[R1] Xu Z, Lin Y S. A stretchable terahertz parabolic-shaped metamaterial[J]. *Advanced Optical*
*Materials*, 2019, 7(19): 1900379.

[R2] Chanda D, Shigeta K, Gupta S, et al. Large-area flexible 3D optical negative index
metamaterial formed by nanotransfer printing[J]. *Nature nanotechnology*, 2011, 6(7): 402-407.

2. The authors claim that the proposed sensor possesses “shows great potential for
future applications in the sensing layer of flexible IoTs and wearable devices.”. In fact,
all data in the manuscript come from ideal test setups (Supplementary Fig. S16).
Besides, it is worth noting that the bulky 2D THz scanning platform must be taken into

*consideration when it is used for practical application. It's hard to imagine how the*
*author could implement the sensor into a wearable device. It is recommended that*
*authors describe the suitable application scenarios in their manuscripts.*

**Response:** Thank you very much for your comments. In the main text, the THz 2D
scanning platform was only used for experimental validation of the 2D strain mapping
capability of our meta-sensor, rather than being considered as a solution for subsequent
applications.

Fig. R4. (a) Display of existing miniaturized THz camera (reported in Ref. [R3]). (b) Display of a
flexible THz camera/scanner (reported in Ref. [R4]).

In fact, great efforts have been made to develop alternative technologies for replacing
the bulky 2D THz scanning platform. The recent advancements in THz imaging
systems show great promise in significantly improving the detection efficiency and
bringing our strain meta-sensor array closer to practical applications. The recent
advancements in THz imaging systems can significantly improve the detection
efficiency and bring our strain meta-sensor array closer to practical applications. For
example, a miniaturized THz camera shown in Fig. R4(a) has been achieved through
“multiple pixels” imaging technology (Ref. [3]), which has a compact size of $5 \times 5 \times 3$
140 cm^3 (Fig. R4(a)) and can quickly detect the amplitude distribution of THz signals across
a plane with the size of $2.9 \times 2.9 \text{ mm}^2$. According to the similar working principle,
flexible THz cameras and wearable scanners have also been proposed in Ref. [R4],
eliminating the need for complex optical components and systems (Fig. R4(b)).

Fig. R5. (a) Illustration of the THz single-pixel imaging. (b) Single pixel imaging system diagram
(reported in Ref. [R6]).

In addition, the emerging THz computational single-pixel imaging technology can
**further compress the size of the THz receiver into a smaller-sized single-pixel** (Fig.
R5(a)) without sacrificing the spectral resolution capabilities of THz-TDS systems
(Refs. [R5-R10]). For example, in Ref. [R6], a single-point THz detector was obtained
to acquire real-time THz images in a $\sim 1.2 \times 1.2 \text{ cm}^2$ area, achieving a resolution of 32
$\times 32$ pixels at a rate of 6 frames-per-second (Fig. R5(b)). With the development of the
155 THz technology, we believe that the corresponding imaging area, as well as the spatial
resolution, will be significantly improved in the near future. Therefore, **once integrated**
**with this technology, the adopted bulky 2D THz scanning platform can be omitted,**
thereby contributing to the rapid realization of our devices from lab to fab.

Regarding the applications in the sensing layer of flexible IoTs and wearable devices
mentioned in our manuscript, we have further extended our design to a reflection type
with the same sensing mechanism and validated it experimentally according to the
suggestion of Reviewer #3 (Supplementary Note S8 and Fig. S13-15). Our reflection
type design allows for isolating the effect of any extra media at the back of the
transmission type, thus making it more practical for real-world applications.

**In conclusion, simultaneously considering great expandability of the proposed**
**strain detection mechanism and the evolving THz imaging technology, our**

**strategy indeed demonstrates huge potential in flexible IoTs and wearable devices.**

**Reference:**

[R3] Al Hadi R, Sherry H, Grzyb J, et al. A 1 k-pixel video camera for 0.7–1.1 terahertz imaging
applications in 65-nm CMOS[J]. IEEE Journal of Solid-State Circuits, 2012, 47(12): 2999-3012.

[R4] Suzuki D, Oda S, Kawano Y. A flexible and wearable terahertz scanner[J]. Nature Photonics,
2016, 10(12): 809-813.

[R5] Chan W L, Charan K, Takhar D, et al. A single-pixel terahertz imaging system based on
compressed sensing[J]. Applied Physics Letters, 2008, 93(12).

[R6] Stantchev R I, Yu X, Blu T, et al. Real-time terahertz imaging with a single-pixel detector[J].
Nature communications, 2020, 11(1): 2535.

[R7] Watts C M, Shrekenhamer D, Montoya J, et al. Terahertz compressive imaging with
metamaterial spatial light modulators[J]. Nature photonics, 2014, 8(8): 605-609.

[R8] Zhao J, E Y, Williams K, et al. Spatial sampling of terahertz fields with sub-wavelength
accuracy via probe-beam encoding[J]. Light: Science & Applications, 2019, 8(1): 55.

[R9] Stantchev R I, Sun B, Hornett S M, et al. Noninvasive, near-field terahertz imaging of hidden
objects using a single-pixel detector[J]. Science advances, 2016, 2(6): e1600190.

[R10] Chen S C, Feng Z, Li J, et al. Ghost spintronic THz-emitter-array microscope[J]. Light:
Science & Applications, 2020, 9(1): 99.

*3. In practical scenarios, particularly in wearable applications they claimed, out-of-*
*plane interference in the z-axis direction is typically inevitable, particularly given the*
*flexibility of their substrate, making them susceptible to such deformations. Such*
*interference may manifest as bending and distortion. Notably, the authors exclusively*
*investigated in-plane deformations. It is advisable for the authors to augment their*
*study by incorporating simulation or test data that considers the sensor's response to*
*bending and twisting deformations.*

**Response:** Thank for your valuable comments. According to your suggestion, we have
evaluated the resonance response of our meta-sensor array under bending and twisting
deformations from the aspect of measurements.

**(1) Bending test**

The schematic diagram of the bending process is shown in Fig. R6(a). Both ends of the
sample were clamped by two movable fixtures, and then by tuning the spacing between
these two fixtures, the bending degree of the sample can be modified continuously
(inset of Fig. R6(a)). The initial length (L) of the sample without bending is 15 mm.

When the sample was gradually bent along the x/y direction, little resonant frequency
 shifting (ED resonance @ x -directional bending; MD resonance @ y -directional
 bending) can be observed in Figs. R6(b-c) (as L decreased from 15 mm to 10 mm),
 denoting that our design is insensitive to the out-of-plane bending deformations.

 Fig. R6. Bending response test for the meta-sensor array. (a) Schematic illustration of the bending
 process and the photograph of the bent meta-sensor sample. THz transmission spectra of the meta-
 sensor in the process of the fixtures gradually moving to the center: (b) for the x -directional bending;
 (c) for the y -directional bending.

(2) Twisting test

When conducting the twisting test, as shown in Fig. R7(a), one end of the sample (with
 the length of 2.5 cm) was fixed, and the other was rotated by $\theta = 0^\circ, 20^\circ, 40^\circ, 60^\circ, 80^\circ,$
 $100^\circ, 120^\circ, 140^\circ, 160^\circ, 180^\circ$ (Fig. R7(b)). The measured THz spectra are presented in
 Figs. R7(c) (rotation along x axis) and 7(d) (rotation along y axis), respectively. When
 the twisting angle gradually increases from 0° to a threshold -- 140° @ rotation along x
 axis and 120° @ rotation along y axis, both the MD and ED resonance peaks
 simultaneously shift towards lower frequencies as well as the transmittance magnitude
 decays, which may be attributed to the orthogonal strains induced by the applied
 rotational deformations. However, once the twisting angle exceeds the corresponding
 thresholds, the changes in the MD and ED resonances become irregular due to the
 sample appearing folded.

According to your valuable suggestion, we have added the above responses of our
 sample to the bending and twisting deformations into the Supporting Information as
 Note S9 and Note S10, respectively.

4. The content provided in their response document was not comprehensively included

in the supplementary materials. It is strongly recommended that the authors integrate

References [1-3] into the Comparison table within the supplementary material.

**Response:** Thank you for your kind reminder. We have integrated the mentioned

references into the Table S1 of the Supporting Information as:

“S18. Xu, Z. & Lin, Y.-S. A Stretchable Terahertz Parabolic-Shaped Metamaterial.

Advanced Optical Materials 7, 1900379 (2019).

S19. Li, J. et al. Flexible terahertz metamaterials for dual-axis strain sensing. Optics

Letters 38, 2104–2106 (2013).

S20. Khatib, O., Tyler, T., Padilla, W. J., Jokerst, N. M. & Everitt, H. O. Mapping active

strain using terahertz metamaterial laminates. APL Photonics 6, 116105 (2021).”

**Reviewer #2 (Remarks to the Author):**

*The authors have done a lot of work to address the review's comments, I think the MS*
*could be accepted for publication in this version.*

**Response:** Thank you very much for your previous comments and suggestions, which
have significantly improved the quality of our manuscript.

**Reviewer #3 (Remarks to the Author):**

*The author's response to question 2 is unsatisfactory. It is evident that the author stated*
*the proposed scheme's application scenario is a wearable device, indicating that only*
*an experimental plan utilizing the reflection system can accomplish this objective.*
*However, the author's reply indicates that the lack of experimental equipment restricts*
*the reflection scheme to simulation only. However, based on my understanding,*
*reflection and transmission geometry are the fundamental optical path structures in*
*THz systems and can be flexibly adapted. Moreover, numerical calculation in real-*
*world for a reflection system is more challenging. If there is only simulation data*
*available without any experimental data, it implies that the proposed scheme may not*
*be feasible for the intended goal or merely suggests that future schemes based on this*
*proposal could be beneficial for future applications.*

**Response:** We sincerely appreciate your comments and apologize for only providing
the simulation results of the reflection-type meta-sensor in our previous response letter
and Supporting Information. According to your valuable suggestion, we have
conducted the stretching experiments to further verify the feasibility of the reflection-
type design shown in Figs. R8(a-b). Prior to carrying out the measurements, we have
successfully rebuilt the optical path of the THz time-domain system to make it suitable
for reflective THz signal detection.

With the help of the new test system, the reflection time-domain signals of the sample
applied with the unidirectional tensile strain were first measured and presented in Figs.
R8(c) (for x -directional strain) and 8(d) (for y -directional strain). Then, the
corresponding spectra information can be obtained (Figs. R8(e) and (h)) through
applying the Fourier transform to the time signals (insets in Figs. R8(c-d)). Similar to
the transmission-type meta-sensor, when the sample is stretched along the x direction,
the measured ED resonance gradually shifts from 0.4211 THz to 0.3640 THz as the
strain increases from 0 to 65% (Fig. R8(e)), which is consistent with the simulation
results (Fig. R8(f)). Conversely, when the strain along the y direction increases from 0
to 70%, the measured MD resonance (Fig. R8(h)) matches well with the simulated one

(Fig. R8(i)), and decreases from 0.3225 THz to 0.2999 THz. Additionally, it is worth
 noting that the remaining resonance (e.g., MD resonance during stretching in the x
 direction, ED resonance during stretching in the y direction) in these two processes
 barely shifts (Figs. R8(g) and (j)). These experimental results have successfully proved
 the unidirectional tensile strain detection ability of the proposed reflective meta-sensor.

 Fig. R8. Unidirectional tensile strain detection performance evaluation of a reflective meta-sensor.
 (a-b) Structure diagram of the reflective meta-sensor. Experimental time-domain spectra of the
 meta-sensor array applied with different tensile strains (varying from 0 to 70% by 5% per step): (c)
 for the x direction; (d) for the y direction. The structural parameters of the proposed meta-sensor are
 $h_1 = 320 \mu\text{m}$, $h_2 = 1000 \mu\text{m}$, $h_3 = 0.2 \mu\text{m}$, $P_x = 320 \mu\text{m}$, $P_y = 320 \mu\text{m}$, $d = 161 \mu\text{m}$. Corresponding
 frequency spectra calculated from the time-domain spectra (insets in (c-d)) through performing
 Fourier Transform: (e) for the x direction; (h) for the y direction. Simulated frequency spectra of the
 sample under different strains: (f) for the x direction; (i) for the y direction. Extracted resonance

information from (e-f) and (h-i) are plotted in (g) and (j). The data in (g) and (j) are presented as
 mean \pm s.d. of $n \geq 6$ independent measurements.

 **Fig. R9.** Bidirectional tensile strain detection performance evaluation of the reflective meta-sensor.
 Experimental reflection spectra of the meta-sensor array simultaneously applied with x - and y -
 directional tensile strain: (a) x -directional tensile strain varying from 5% to 30% while the
 orthometric strain is maintained at 5%, 10%, 15%, 20%, 25%, and 30%; (b) y -directional tensile
 strain varying from 5% to 30% while the orthometric strain is maintained at 5%, 10%, 15%, 20%,
 25%, and 30%. Extracted resonance information from (a) and (b) are plotted in (c) and (d). The data
 in (c-d) are presented as mean \pm s.d. of $n \geq 6$ independent measurements.

In addition, bidirectional tensile strain detection performance of the proposed reflective
meta-sensor has been evaluated. As shown in Fig. R9(a), with the x -directional strain
varying from 5% to 30% by 5% per step, the strain applied in the y -direction is fixed at
5%, 10%, 15%, 20%, 25% and 30%. In such cases, the ED resonance gradually shifts
to a lower frequency as the x -directional strain increases, while the MD resonance
always remains localized at a certain value only associated with the y -directional
deformation ratio (5% @ 0.3212 THz, 10% @ 0.3183 THz, 15% @ 0.3165 THz, 20%
@ 0.3143 THz, 25% @ 0.3126 THz, and 30% @ 0.3116 THz). By switching the strain
loading, the opposite phenomenon can be revealed in Fig. R9(b), i.e., the MD resonance
frequency decreases with increasing strain in the y -direction, while the ED resonance
always stays at a specific value, which is only related to the x -directional deformation
ratio (5% @ 0.4173 THz, 10% @ 0.4148 THz, 15% @ 0.4107 THz, 20% @ 0.4072
326 THz, 25% @ 0.4018 THz, and 30% @ 0.3970 THz). These results demonstrate that the
327 reflective meta-sensor also favours independent and noninterfering monitoring of the
328 orthogonal strains. Additionally, the experimental results agree well with the simulated
results shown in Figs. R10(a-b), further proving the veracity of our strategy.
Furthermore, in Figs. R9(c-d), the dynamic ED and MD resonance frequencies during
stretching have been extracted from Figs. R9(a-b) to directly exhibit the corresponding
relation between the bidirectional strains applied to our sensor and its resonance shifting.
**All the experimental results in Figs. R8-9 indicate that the sensing mechanism we**
**proposed can be directly extended to the design of reflective strain sensor elements,**
**making it more suitable for practical applications.**

The abovementioned description and Figs. R8-10 have been added into the Supporting
Information as Note S8 and Fig. S13-15 to replace the previous theoretical model under
ideal conditions -- vacuum is used as the substrate to reveal the working mechanism of
the reflection-type meta-sensor.

Fig. R10. Simulated spectra of the reflective meta-sensor array simultaneously applied with x - and
 y -directional tensile strain: (a) x -directional tensile strain varying from 5% to 30% while the
 orthometric strain is maintained at 5%, 10%, 15%, 20%, 25%, and 30%; (b) y -directional tensile
 strain varying from 5% to 30% while the orthometric strain is maintained at 5%, 10%, 15%, 20%,
 25%, and 30%.

**Reviewer #4 (Remarks to the Author):**

*The authors took into account my comments. In my opinion, the article has become*
*better and can be published in the journal Nature Communications.*

**Response:** Thank you for your recognition. And your constructive comments and
suggestions have undoubtedly contributed to the improvement of our manuscript.

REVIEWERS' COMMENTS

Reviewer #1 (Remarks to the Author):

The authors succeeded in clarifying all the points raised in this revision. It is recommended that this manuscript be considered for publication as is.

**Point-by-point response to reviewer:**

**Reviewer #1 (Remarks to the Author):**

*The authors succeeded in clarifying all the points raised in this revision. It is*
*recommended that this manuscript be considered for publication as is.*

**Response:** Thank you very much for your previous comments and suggestions, which
have significantly improved the quality of our manuscript.
